# $\alpha$-MDF: An Attention-based Multimodal Differentiable Filter for Robot State Estimation

**Xiao Liu[1], Yifan Zhou[1], Shuhei Ikemoto[2], and Heni Ben Amor[1]**

[1]Interactive Robotics Lab, Arizona State University

[2]Kyushu Institute of Technology

[1]{xliu330,yzhou298,hbenamor}@asu.edu  [2]ikemoto@brain.kyutech.ac.jp

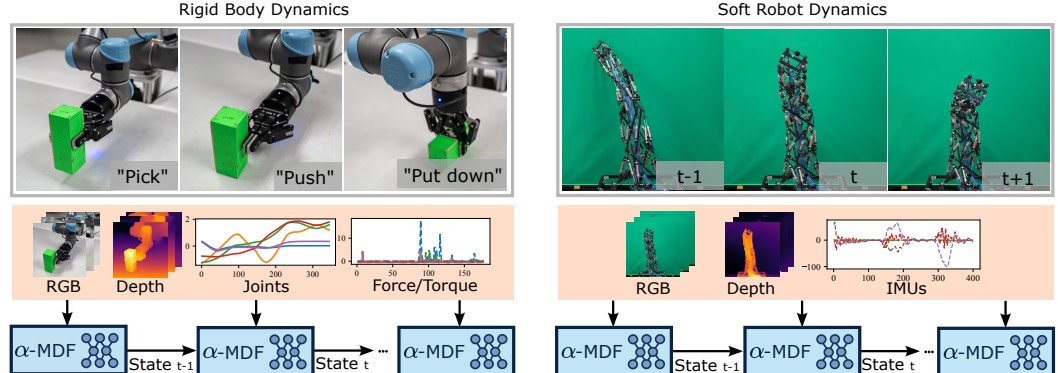

Figure 1: The attention-based Multimodal Differentiable Filter ($\alpha$-MDF) framework enables robot state estimation in multimodal settings, applicable to both rigid body robots and soft robots.

**Abstract:** Differentiable Filters are recursive Bayesian estimators that derive the state transition and measurement models from data alone. Their data-driven nature eschews the need for explicit analytical models, while remaining algorithmic components of the filtering process intact. As a result, the gain mechanism – a critical component of the filtering process – remains non-differentiable and cannot be adjusted to the specific nature of the task or context. In this paper, we propose an attention-based Multimodal Differentiable Filter ($\alpha$-MDF) which utilizes modern attention mechanisms to learn multimodal latent representations. Unlike previous differentiable filter frameworks, $\alpha$-MDF substitutes the traditional gain, e.g., the Kalman gain, with a neural attention mechanism. The approach generates specialized, context-dependent gains that can effectively combine multiple input modalities and observed variables. We validate $\alpha$-MDF on a diverse set of robot state estimation tasks in real world and simulation. Our results show $\alpha$-MDF achieves significant reductions in state estimation errors, demonstrating nearly 4-fold improvements compared to state-of-the-art sensor fusion strategies for rigid body robots. Additionally, the $\alpha$-MDF consistently outperforms differentiable filter baselines by up to 45% in soft robotics tasks. The project is available at `alpha-mdf.github.io` and the codebase is at `github.com/ir-lab/alpha-MDF`

**Keywords:** Differentiable Filters, Sensor Fusion, Multimodal Learning.

## 1 Introduction

Recursive Bayesian filters, in particular Kalman filters, are a core component of many robotic and autonomous systems [1]. These filters offer a probabilistic framework that enables effective state estimation and allowing robots to perceive and respond to dynamic environmental conditions [2, 3, 4].

7th Conference on Robot Learning (CoRL 2023), Atlanta, USA.

Constructing the analytical models and characterizing their noise profiles can be an overwhelming undertaking and requires supplementary measures, e.g., system identification [5]. In addition, scalability still poses a significant obstacle, particularly when dealing with non-linear and high-dimensional systems. Advanced techniques such as the ensemble Kalman filter [6] have been developed to tackle this challenge. However, they may still require careful manual design of the data pipeline and filtering process, especially in the presence of multimodal data sources. A potential alternative methodology is to derive the underlying models for filtering from data alone. Recent advancements in Deep state-space models (DSSMs) [7] provide effective solutions for understanding the state and measurement estimation from observed sequences as data-driven approaches [7, 8, 9]. Such approaches do not need to derive explicit system dynamics, which is essential and challenging in traditional filtering techniques. A subclass of algorithms derived from DSSMs, called Differentiable Filters (DFs), focus on learning state transition and measurement models from data while retaining the fundamental principles of Bayesian recursive filtering. This combination of properties renders DFs particularly well-suited for systems with complex dynamics and diverse sensor observations.

In this paper, we introduce a novel class of differentiable filters built upon neural attention mechanisms. The key innovation lies in the substitution of the traditional Kalman gain with an attention mechanism for filtering with multimodal observations. This approach allows for the learning of a highly specialized and task-specific gain mechanism. The utilization of multimodal observations, also known as multimodal learning [10], has shown substantial advantages in various robotics applications [11, 12, 13]. By harnessing information from diverse modalities such as vision [14, 15, 16], language [17, 18, 19, 20], and tactile sensing [21, 22], robots can learn to better interpret their surroundings, produce more accurate estimates of their own internal state, and consequently improve the overall decision-making process. We propose the attention-based Multimodal Differentiable Filters ($\alpha$-MDF) framework, as shown in Fig. 2, each module is learnable and operates in latent space. The primary contributions are: (1) **Attention Gain**: Our approach is an attention-based strategy that replaces the conventional Kalman gain in the measurement update step, as depicted by the colored blocks in Fig. 2. The gain mechanism is learned to update the current state based on multimodal observations. (2) **Latent Space Filtering**: our proposed differentiable framework operates in a latent space, learning high-dimensional representations of system dynamics and capturing intricate nonlinear relationships. This approach proves particularly advantageous for highly nonlinear systems. (3) **Empirical evaluations**: $\alpha$-MDF achieves significant reductions in state estimation errors, demonstrating nearly 4-fold improvements compared to state-of-the-art sensor fusion strategies in multimodal manipulation tasks. Furthermore, $\alpha$-MDF accurately models the non-linear dynamics of soft robots, consistently surpassing differentiable filter baselines by up to 45%.

## 2  Related Work

Differentiable Filters (DFs) have garnered attention as learnable non-linear state-space models [23, 24, 25]. Previous works [26, 27] have integrated neural network components into robotic algorithms, such as BackpropKF, which combines backpropagation with neural networks to train Kalman Filters. Similarly, research in Differentiable Particle Filters (DPFs) [28, 29] has also leveraged learnable modules to address the challenges of filtering and state tracking. Algorithmic priors have been utilized to improve the learning efficiency of DPFs, and adversarial methods have been used for posterior estimation [30]. However, the gain mechanism in the traditional Kalman filter is not differentiable and has not been incorporated into the learning procedure of the DFs mentioned earlier. In a recent study [9], the effectiveness of DFs in training and modeling uncertainty with noise profiles has been demonstrated. Typically, multi-layer perceptrons integrated with an RNN layer are employed in the implementation of DFs. Performance on real-world tasks has shown considerable improvement in state tracking accuracy [9, 22, 24, 31, 32], and results indicate that the adoption of end-to-end learning is crucial for accurately learning noise models. However, as noted in [8], the use of RNN has been shown to be "a limiting factor for learning accurate models" and may "lead to a non-Markovian state-space". Furthermore, the traditional Kalman gain for DFs in [9, 22, 24] remains non-learnable, despite the numerous advancements made thus far in differentiable filters

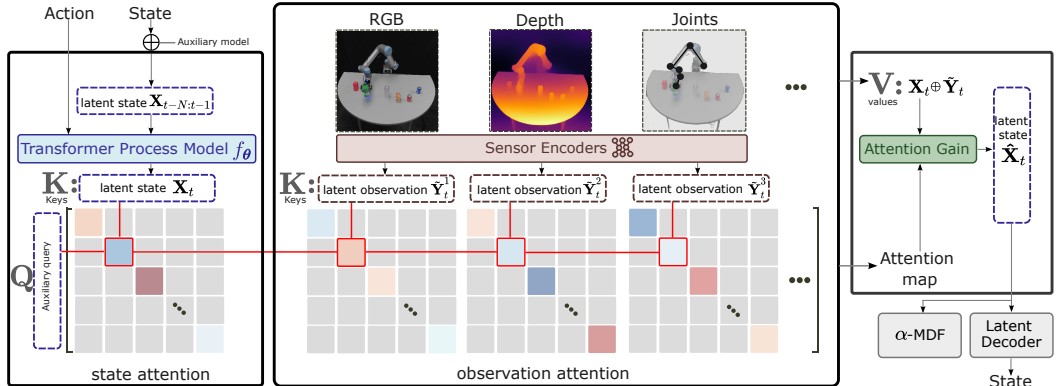

Figure 2: The $\alpha$-MDF framework consists of a transformer process model, sensor encoders, and an attention gain update model. The transformer process model predicts the following latent state, while the sensor encoders learn latent representations from observations. The attention gain model then corrects the predicted latent state using the learned representations.

and particle filters. While DFs have demonstrated considerable promise as differentiable sequential models, their application in multimodal settings has thus far received limited attention. One notable exception is the recent work in [22], which proposes sensor fusion strategies associated with DFs. Employing multimodal environments often necessitates the adoption of representation learning methodologies [33], which require learning a latent representation for capturing intricate static and dynamic features [8, 12, 34]. Studies such as [11] have highlighted the effectiveness of blending networks in learning a shared representation based on Conditional Neural Processes [35]. Other studies, including [21, 12], have adopted a self-supervised approach involving variational autoencoders (VAE) [36] to discover latent representations for stable manipulation policies. The majority of multimodal latent representations are obtained through policy learning. However, in certain situations, a gating technique [37] that focuses on specific modalities can be employed to enhance policy robustness. In light of this, we propose an alternative approach which utilizes modern attention mechanisms to learn multimodal latent representations *at both the level of modalities and observations*. The novel approach fuses multiple types of modalities and variables in a *dynamic, context-dependent* fashion, thereby enabling synergies between their respective qualities, structures, relevance, and degrees of redundancy.

## 3   Multimodal Differentiable Filters

We introduce a novel approach called **a**ttention-based **M**ultimodal **D**ifferentiable **F**ilters ($\alpha$-**MDF**), which combines differentiable filters with insights from transformer models. This approach performs state estimation and fusion of multiple sensor modalities in a unified and differentiable manner. We first discuss Recursive Bayesian filtering as the general technique used to estimate the state $\mathbf{x}_t$ of a discrete-time dynamical system. Thereafter, we provide the details of our specific algorithm. Given a sequence of actions $\mathbf{a}_{1:t}$ and noisy observations $\mathbf{y}_{1:t}$, the posterior distribution of the state can be represented by the following equation:

$$p(\mathbf{x}_t | \mathbf{a}_{1:t}, \mathbf{y}_{1:t}, \mathbf{x}_{1:t-1}) \propto p(\mathbf{y}_t | \mathbf{a}_t, \mathbf{x}_t)\, p(\mathbf{x}_t | \mathbf{a}_{1:t-1}, \mathbf{y}_{1:t-1}, \mathbf{x}_{1:t-1}). \tag{1}$$

We can denote the belief of the state as $\mathrm{bel}(\mathbf{x}_t) = p(\mathbf{x}_t | \mathbf{a}_{1:t}, \mathbf{y}_{1:t}, \mathbf{x}_{1:t-1})$. Assuming the Markov property, where the next state is dependent only on the current state, we get the following expression:

$$\mathrm{bel}(\mathbf{x}_t) = \eta \underbrace{p(\mathbf{y}_t | \mathbf{x}_t)}_{\text{observation model}} \prod_{t=1}^{t} \overbrace{p(\mathbf{x}_t | \mathbf{a}_t, \mathbf{x}_{t-1})}^{\text{state transition model}} \mathrm{bel}(\mathbf{x}_{t-1}), \tag{2}$$

where $\eta$ is a normalization factor, $p(\mathbf{y}_t | \mathbf{x}_t)$ is the observation model and $p(\mathbf{x}_t | \mathbf{a}_t, \mathbf{x}_{t-1})$ is the transition model. The transition model describes the laws that govern the evolution of the system state, while the observation model identifies the relationship between the hidden, internal state of the system and observed, noisy measurements.

## 3.1 $\alpha$-MDF

We utilize an ensemble method for Bayesian filtering wherein each ensemble member represents a compact robot state. Figure 2 shows the procedural steps of how this compact representation, known as the latent state, is obtained and get updated. The filtering process includes two essential steps, namely *prediction* and *update*, both of which are also implemented through neural networks. Most importantly, we replace the Kalman gain step with an attention mechanism, which is trained to weigh observations against predictions based on the current context. Additionally, we demonstrate that attention can be used to balance and weigh different modalities, e.g., video, depth, inertial measurements, against each other. We will see that both steps can be naturally integrated into a single **attention gain** (AG) module.

Let $\mathbf{X}_{0:N}$ denote the latent states with dimension $d_x$ of $N$ steps in $t$ with number of $E$ ensemble members, $\mathbf{X}_{0:N} = [\mathbf{x}_{0:N}^1, \ldots, \mathbf{x}_{0:N}^E]$, where $E \in \mathbb{Z}^+$.

**Prediction step**: In this step, the state transition model takes the previous states with the current action, and predicts the next state. To this end, we leverage the capabilities of transformer-style neural networks [38]. In addition, we generate a probability distribution over the posterior by implementing the state transition model as a stochastic neural network. Therefore, we can use the following prediction step to update each ensemble member, given a sequence of latent states $\mathbf{X}_{t-N:t-1}$:

$$\mathbf{x}_{t|t-N:t-1}^i \sim f_{\boldsymbol{\theta}}(\mathbf{x}_{t|t-N:t-1}^i | \mathbf{a}_t, \mathbf{x}_{t-N:t-1}^i), \ \forall i \in E. \tag{3}$$

Where $f_{\boldsymbol{\theta}}$ is a transformer-style neural network with multiple attention layers. In our framework, the latent state and the action at $t$ are processed by positional and type embedding layers [38] prior to being fed into $f_{\boldsymbol{\theta}}$. Matrix $\mathbf{X}_{t|t-N:t-1} \in \mathbb{R}^{d_x \times E}$ holds the updated ensemble members which are propagated one step forward in latent space. For simplicity, we represent $\mathbf{X}_{t|t-N:t-1}$ as $\mathbf{X}_t$ to denote the predicted state. Further elaboration on positional embeddings, type embeddings, and filter initialization can be found in Appendix A.1 for more comprehensive details.

**Update step**: A crucial step of the filtering process is the update step, which involves calculating the gain value. Traditional KF uses the Kalman gain to correct the state by comparing the uncertainty or covariance obtained from state space and observation space, it requires an explicit function to map the state to the measurement. As a result, some sensor measurements like images or deep-learned features are unable to be used in the formulation directly. The proposed **attention gain** (AG) module, on the other hand, eliminates the

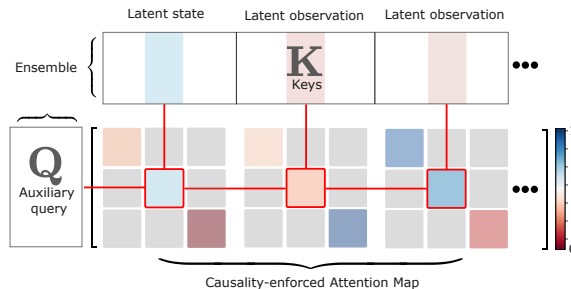

Figure 3: Attention gain (AG) module uses a learned causality-enforced attention map to replace Kalman gain.

need for an explicit observation model and can directly utilize high-dimensional features. By leveraging this approach, our framework enables a more flexible and efficient integration of measurements without the explicit requirement of a mapping function from the state to the measurement domain. Instead of using one sensor encoder, we use multiple sensor encoders $[s^1(\cdot), s^2(\cdot), \cdots, s^M(\cdot)]$ to learn latent observations from each modality:

$$\tilde{\mathbf{y}}_t^{(i,m)} \sim s^m(\tilde{\mathbf{y}}_t^{(i,m)} | \mathbf{y}_t^m), \ \forall \ i \in E, \ m \in M. \tag{4}$$

$M$ is the number of modalities in the system, $M \in \mathbb{Z}^+$. The encoders generate a series of latent observations, $\tilde{\mathbf{Y}}_t = [\tilde{\mathbf{Y}}_t^1, \cdots, \tilde{\mathbf{Y}}_t^M] \in \mathbb{R}^{Md_x \times E}$, where $\tilde{\mathbf{Y}}_t^m = [\tilde{\mathbf{y}}_t^{(1,m)}, \cdots, \tilde{\mathbf{y}}_t^{(E,m)}] \in \mathbb{R}^{d_x \times E}$. The latent observations are then concatenated with predicted state $\mathbf{X}_t$ as input to the AG model:

$$\hat{\mathbf{X}}_t = \text{softmax}\left(\frac{\boldsymbol{Q}(\mathbf{X}_t' \oplus \tilde{\mathbf{Y}}_t')^T}{\sqrt{E}} \circ \tilde{\boldsymbol{M}}\right)(\mathbf{X}_t \oplus \tilde{\mathbf{Y}}_t), \tag{5}$$

where "⊕" denotes the concatenation and "∘" is the Hadamard product, and $\hat{\mathbf{X}}_t$ is the final output. In general, an attention module typically receives three sequences of tokens: queries $\boldsymbol{Q}$, keys $\boldsymbol{K}$ and values $\boldsymbol{V}$. In our case, we define $(\mathbf{X}'_t \oplus \tilde{\mathbf{Y}}'_t)$ as the $\boldsymbol{K}$ tokens, where $\mathbf{X}'_t$ and $\tilde{\mathbf{Y}}'_t$ are obtained by zero-centering, and the actual values of $(\mathbf{X}_t \oplus \tilde{\mathbf{Y}}_t)$ are regarded as the $\boldsymbol{V}$ tokens. As illustrated in Fig. 3, the length of the $\boldsymbol{K}$ tokens is denoted as $d_k = (M+1)d_x$, where each token has a dimension of $E$, representing the distribution along this particular token index.

In a traditional attention mechanism, the proximity of $\boldsymbol{Q}$ and $\boldsymbol{K}$ is measured, and $\boldsymbol{V}$ that is associated with $\boldsymbol{K}$ is utilized to generate outputs. However, we posit that within each latent vector, every index is probabilistically independent, and index $i$ of a latent state should only consider index $i$ of each latent observation. To accomplish this, we utilize matrix $\tilde{\boldsymbol{M}}$ to retain only the diagonal elements of each $(d_x \times d_x)$ attention map, which enforces causality and allows the attention weights to be determined according to the corresponding indices. As depicted in Fig. 3, the red line represents the mapping for a single latent state token index. Auxiliary query tokens $\boldsymbol{Q} \in \mathbb{R}^{d_x \times E}$ are introduced as trainable parameters in the neural network to facilitate learning. It is important to note that both the $\boldsymbol{Q}$ and $\boldsymbol{K}$ tokens undergo positional embedding before being fed into the AG module.

**Placing Conditions on the Latent Space:** Within the framework of Kalman filters, the update step plays a crucial role in aligning the predicted observation with the observations obtained from sensors. Within the framework of $\alpha$-MDF, we ensure consistency in the latent space by introducing a decoder model $\mathcal{D}$. This decoder model, implemented using multilayer perceptrons, projects the latent space onto the actual state space. By doing so, we resolve the alignment challenges in multimodal learning [39], and gain meaningful comparisons when conducting sensor fusion and measurement update. Let $\boldsymbol{x}_t$ be the ground truth state at $t$, the loss functions are defined as:

$$\mathcal{L}_{f_{\boldsymbol{\theta}}} = \|\mathcal{D}(f_{\boldsymbol{\theta}}(\mathbf{X}_t)) - \boldsymbol{x}_t\|_2^2, \quad \mathcal{L}_{\text{e2e}} = \|\mathcal{D}(\hat{\mathbf{X}}_t) - \boldsymbol{x}_t\|_2^2, \quad \mathcal{L}_s = \|\mathcal{D}(s^m(\mathbf{y}_t^m)) - \boldsymbol{x}_t\|_2^2. \quad (6)$$

The final loss function is $\mathcal{L} = \mathcal{L}_{f_{\boldsymbol{\theta}}} + \mathcal{L}_{\text{e2e}} + \mathcal{L}_s$, where $\mathcal{L}_{\text{e2e}}$ is the end-to-end loss. $\mathcal{L}_{f_{\boldsymbol{\theta}}}$ is used to supervise the state transition model. The latent observation conditioning is provided with $\mathcal{L}_s$ during the training process, note that the conditioning operation is applied when the modalities collectively provide information about the complete state. The modular architecture of $\alpha$-MDF provides a key advantage in facilitating training and testing with masked modalities. The attention matrix $\tilde{\boldsymbol{M}}$ can be disabled (set the attention values to zero) based on different input sensor modalities, thus improving the model's resilience to missing modalities.

## 4 Experiments

We conduct a series of experiments to evaluate the efficacy of the $\alpha$-MDF framework. Specifically, we aim to answer the following questions: (a) Can the $\alpha$-MDF framework generalize across various tasks? (b) To what extent does the new filtering mechanism improve state tracking performance when compared to the current state-of-the-art? (c) How does the use of multiple modalities compare to a subset of modalities for state estimation with differentiable filters? Therefore, we evaluate the effectiveness of $\alpha$-MDF across multiple robotics tasks, each with distinct setups: (1) **Visual odometry** for autonomous driving, (2) **Robot manipulation** employing multi-modalities in both real-world and simulation, and (3) **Soft robot modeling** task. Our study examines two categories of baselines: (a) DF baselines such as those proposed in [9, 28, 26], including dEKF [9], DPF [28], and dPF-M-lrn [9]; and (b) sensor fusion baselines proposed in [22]. Additional details on the baselines can be found in Appendix A.3.

### 4.1 Visual Odometry Task

In this experiment, we evaluate the performance of $\alpha$-MDF on the popular KITTI Visual Odometry dataset [40]. Since the visual odometry task uses a single modality, we only consider RGB images as the input modality in order to make a fair comparison with the baselines [9, 28, 26]. The actual state is defined as a 5-dimensional vector $\boldsymbol{x} = [x, y, \theta, v, \dot{\theta}]^T$, including the position and orientation of the vehicle, and the linear and angular velocity. We use the latent state $\mathbf{x} \in \mathbb{R}^{256}$ for $\alpha$-MDF. In comparison to dEKF, DPF, and dPF-M-lrn, we observe a reduction in the translational error of

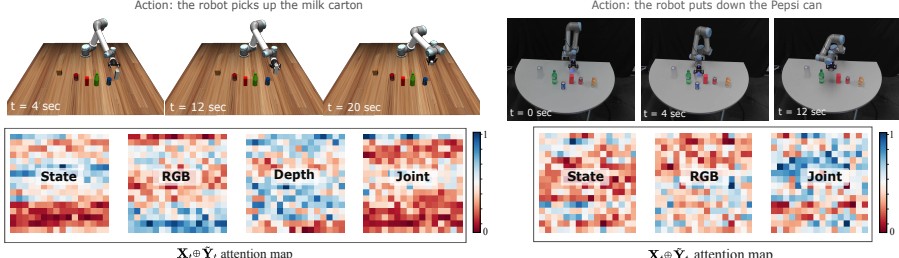

Figure 4: Learned attention gain. **Left**: manipulation in a simulated environment with modalities $[\mathbf{y}^1, \mathbf{y}^2, \mathbf{y}^3]$, and **right**: real robot manipulation with modalities $[\mathbf{y}^1, \mathbf{y}^3]$. The attention maps indicate the attention weights assigned to each modality during model inference. In the visualization, regions in red correspond to low attention values, while those in blue indicate high attention values.

approximately 88%, 83%, and 79% for Test 100/200/400/800. The results also reflect a considerable reduction in rotational error of approximately 64%, 54%, and 46% as compared to each of the baselines. We report a detailed experimental setup and thorough results in Appendix B.1.

## 4.2 Multimodal Manipulation Task

This experiment aims to evaluate the effectiveness of the $\alpha$-MDF framework in a robot manipulation scenario. Specifically, we use $\alpha$-MDF for monitoring the state of a UR5 robot during tabletop arrangement tasks. Similar to behavioral cloning from observation tasks [41], actions are not available as inputs for this study. Instead, we train $\alpha$-MDF to learn how to propagate the state of the robot over time. The evaluation involves three manipulation tasks, namely: (1) estimating the state of the robot in a simulated environment, (2) estimating the state of the real-world robot, and (3) estimating the joint state of the robot and the object being manipulated.

**Task Setup and Data:** For $\alpha$-MDF, we define the latent state $\mathbf{x} \in \mathbb{R}^{256}$ for all the sub-tasks. The actual state of the UR5 robot is defined by $\boldsymbol{x}_R$, which consists of the joint angles ($J_1$-$J_7$) and the Cartesian coordinates $(x, y, z)$ of the robot's end-effector (EE). $\boldsymbol{x}_O$ denotes the state of the object, which only includes the location $(x, y, z)$ of the object. The complete set of modalities comprises $[\mathbf{y}^1, \mathbf{y}^2, \mathbf{y}^3, \mathbf{y}^4]$, where $\mathbf{y}^1 \in \mathbb{R}^{224 \times 224 \times 3}$ represents RGB images, $\mathbf{y}^2 \in \mathbb{R}^{224 \times 224}$ represents depth maps, $\mathbf{y}^3 \in \mathbb{R}^7$ represents proprioceptive inputs (joint angles), and $\mathbf{y}^4 \in \mathbb{R}^6$ represents Force/torque (F/T) sensor readings. However, the input modalities for each of the three tasks may differ; for task (1), it involves $[\mathbf{y}^1, \mathbf{y}^2, \mathbf{y}^3]$, for task (2), it comprises $[\mathbf{y}^1, \mathbf{y}^3]$, while task (3) has $[\mathbf{y}^1, \mathbf{y}^2, \mathbf{y}^3, \mathbf{y}^4]$. A more detailed description of the task setup and data collection is supplied in Appendix B.2.

**Results:** Using the same comparison protocol as in [9, 28, 26], Table 1 compares the proposed framework's performance with other DF baselines. Note that all baselines perform tracking in actual space, therefore, we use a pretrained sensor encoder to process RGB modality for all DFs and supplied the latent embedding to $\alpha$-MDF, as DF base-

Table 1: Result evaluations on UR5 manipulation task

| Method | Real-world (MAE) | | Simulation (MAE) | |
|---|---|---|---|---|
| | Joint (deg) | EE (cm) | Joint (deg) | EE (cm) |
| dEKF [9] | 16.08±0.1 | 5.67±0.1 | 4.93±0.2 | 1.91±0.1 |
| DPF [28] | 15.93±0.1 | 5.08±0.3 | 4.46±0.2 | 1.51±0.2 |
| dPF-M-lrn [9] | 12.83±0.1 | 3.95±0.4 | 3.82±0.2 | 1.26±0.1 |
| $\alpha$-MDF | **7.49±0.1** | **3.81±0.2** | **2.84±0.1** | **1.06±0.1** |

Means±standard errors.

lines only take one modality. $\alpha$-MDF outperforms dEKF and DPF, reducing errors by 33% and 25% in real-world and 45% and 30% in simulation, with an average MAE of 3.81cm and a deviation of 1.06cm from ground truth for end-effector positions. Additionally, $\alpha$-MDF exhibits a 42% and 26% improvement in estimating joint angles compared to dPF-M-lrn. In the case of filtering with *multiple modalities*, results presented in Table 2 show clear improvements achieved by $\alpha$-MDF in comparison to other sensor fusion techniques. The baselines are reproduced following the procedure of [22] by providing the same pretrained sensor encoder to each modality. $\alpha$-MDF outperforms all other methods across all three manipulation tasks. In particular, it cuts the positional error of the end-effector (EE) in half when compared to the crossmodal fusion strategy [22]

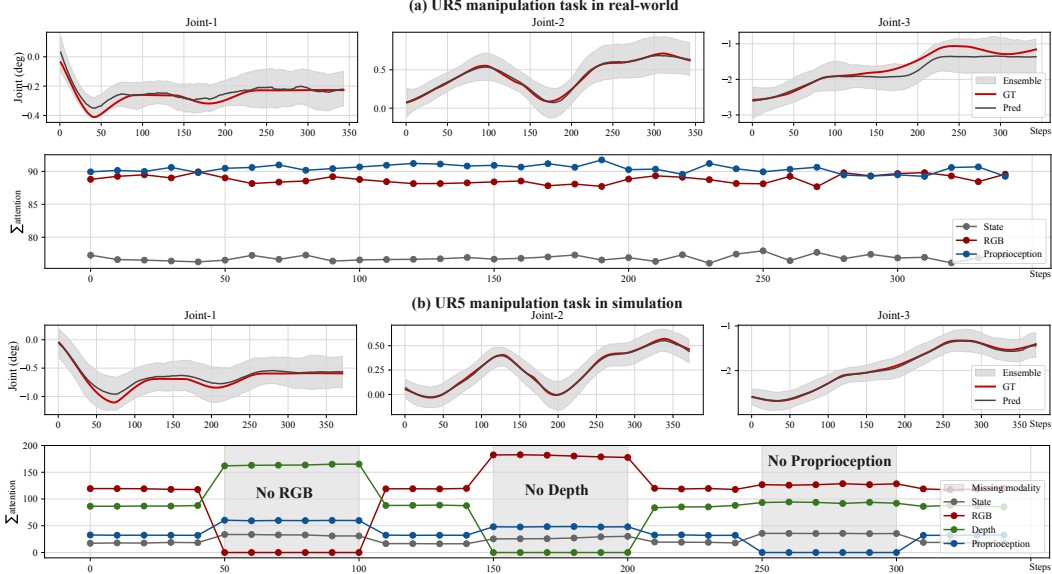

Figure 5: Predicted joint angle trajectories and the corresponding accumulated attention values for each modality. (a) represents the results attained from the actual robot, whereas (b) illustrates attention values for all modalities both with and without masking certain modalities.

Table 2: Result evaluations on UR5 manipulation task with multimodal sensor fusion baselines.

| Method | Simulation (MAE) | | Real-world (MAE) | | Simulation with F/T (MAE) | | |
|---|---|---|---|---|---|---|---|
| | Joint (deg) | EE (cm) | Joint (deg) | EE (cm) | Joint (deg) | EE (cm) | Obj (cm) |
| Feature Fusion [22] | 7.58±0.12 | 3.15±0.16 | 11.25±1.17 | 5.65±0.01 | 3.62±0.09 | 2.72±0.02 | 8.36±0.06 |
| Unimodal [22] | 7.46±0.32 | 3.18±0.03 | 11.02±0.08 | 9.52±0.07 | 3.97±0.08 | 3.63±0.05 | 10.23±0.10 |
| Crossmodal [22] | 3.64±0.34 | 1.91±0.04 | 5.98±0.08 | 7.35±0.05 | 3.12±0.02 | 3.25±0.02 | 5.54±0.02 |
| α-MDF | **2.19±0.09** | **0.75±0.01** | **5.24±0.04** | **3.04±0.01** | **1.41±0.04** | **0.90±0.01** | **1.65±0.01** |

Means±standard errors.

on the real-robot (7.35cm → 3.04cm). In simulation tasks, it achieves an even better reduction in tracking error (3.25cm → 0.90cm in simulation with F/T sensing). We present a visualization of the learned attention gain in the filtering process (Fig. 4) and state tracking results with and without certain modalities (Fig. 5). Despite the attention values changing when certain modalities are missing, α-MDF still achieves stable results. Further results and explanations can be found in AppendixB.2.

### 4.3 Soft robot Modeling

Tensegrity structures [42] have become popular in recent years since they bridge the gap between an inherently flexible system and the ability to use rigid components [43, 44, 45]. However, modeling such a complex system continues to pose considerable challenges due to the high nonlinearity. This experiment involves implementing the α-MDF to model the dynamics of a soft robot system, especially Tensegrity robot [45].

**Task Setup and Data:** The robot structure is shown in Fig. 6 with 5 layers of tensegrity. The actual state of a soft robot at time $t$ is represented by a 7-dimensional vector $\boldsymbol{x}_t = [x, y, z, \mathbf{q}_x, \mathbf{q}_y, \mathbf{q}_z, \mathbf{q}_w]^T$, which denotes the position and orientation of the robot's hand tip. The quaternion vector $\mathbf{q}$ represents the posture of the robot w.r.t the base (layer 1's bottom). In this task, we define $\mathbf{x} \in \mathbb{R}^{256}$ as the latent state. The complete set of modalities comprises

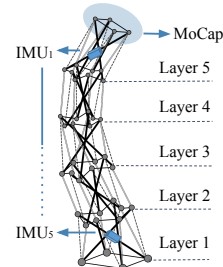

Figure 6: The tensegrity robot structure.

$[\mathbf{y}^1, \mathbf{y}^2, \mathbf{y}^3]$, where $\mathbf{y}^1 \in \mathbb{R}^{224 \times 224 \times 3}$ represents RGB images, $\mathbf{y}^2 \in \mathbb{R}^{224 \times 224}$ is depth maps, and $\mathbf{y}^3 \in \mathbb{R}^{30}$ is proprioceptive inputs (IMUs). The action $\mathbf{a}_t$ of the system is the pressure vector of the 40 pneumatic cylinder actuators, where $\mathbf{a}_t \in \mathbb{R}^{40}$. In this experiment, synthetic depth maps are generated offline using the DPT model [46]. Figure. 7 shows the recordings of RGB and the depth modalities, further details regarding the task setup and data collection is in Appendix B.3.

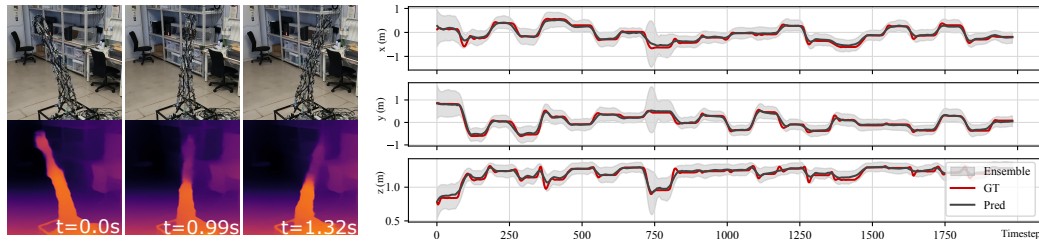

Figure 7: Estimated end-effector (EE) positions for tensegrity robot. **Left**: the RGB and depth modalities $[\mathbf{y}^1, \mathbf{y}^2]$, and **right**: state estimation results with ensemble distribution.

**Results:** The soft robot modeling task is evaluated using 10-fold cross-validation and the mean absolute error (MAE) metric, and the results are presented in Table 3. Our results demonstrate that $\alpha$-MDF outperforms the state-of-the-art methods in terms of DFs, achieving a MAE of 8.99cm. Specifically, our approach yields an MAE on the end-effector (EE) position estimation that is 45%, 34%, and 29% lower than that obtained by dEKF, DPF, and dPF-M-lrn, respectively.

Of the sensor fusion baselines, cross-modal fusion [22] exhibits marginally better outcomes than others, although it do not show any advantages over $\alpha$-MDF in predicting EE positions (2.14cm→1.67cm). Notably, $\alpha$-MDF surpasses the feature fusion strategy by a significant margin of 4-fold. Additionally, appendix B.3 delves into an exploration of the potential benefits of modality selection for state estimation, where

Table 3: Result evaluation on soft robot modeling task.

|  | RGB | Depth | IMUs | EE (cm) | $\mathbf{q}\ (10^1)$ |
|---|---|---|---|---|---|
| dEKF [9] |  |  | ✓ | 16.38±0.10 | 1.01±0.03 |
| DPF [28] |  |  | ✓ | 13.68±0.02 | 0.96±0.03 |
| dPF-M-lrn [9] |  |  | ✓ | 12.66±0.09 | 1.10±0.03 |
| $\alpha$-MDF |  |  | ✓ | **8.99±0.02** | **0.79±0.03** |
| Feature Fusion [22] | ✓ | ✓ | ✓ | 8.35±0.22 | 0.60±0.03 |
| Unimodal [22] | ✓ | ✓ | ✓ | 2.78±0.05 | 0.25±0.02 |
| Crossmodal [22] | ✓ | ✓ | ✓ | 2.14±0.05 | 0.15±0.02 |
| $\alpha$-MDF | ✓ | ✓ | ✓ | **1.67±0.09** | **0.12±0.01** |

Means±standard errors.

optimal combinations can be selected to achieve even higher accuracy. The results presented in Fig. 7 demonstrate the efficacy of $\alpha$-MDF in accurately estimating the state of soft robots in a multimodal setting, the ensemble distribution is indicated by gray shade representing the model uncertainty. With stable performance achieved over an extended duration of inference, $\alpha$-MDF has shown the potential in modeling dynamics for various complex non-linear systems.

## 5   Conclusion

This paper illustrates how utilizing attention as a gain mechanism in differentiable Bayesian filtering and multimodal learning can significantly enhance the accuracy of robot state estimation in numerous tasks. Proposed $\alpha$-MDF is a unique differentiable filter that conducts filtering on a compressed multimodal latent representation, while preserving the integrity of the Kalman filter algorithm component. Our experiments demonstrate that $\alpha$-MDF is appropriate for learning both rigid body and soft robot dynamics, exceeding baseline performance by up to 4-fold. Moving forward, we plan to investigate the value of incorporating additional modalities, such as sound, temperature, and proximity sensing, into $\alpha$-MDF.

**Limitation:** An obvious difference of $\alpha$-MDF when compared to traditional filters is the required learning process – this typically takes multiple hours of training on current machines. In a similar vain, an inherent assumption is that the training and test distributions do not differ substantially, i.e., the problem of concept drift. To date, we have successfully tested the algorithm with latent states consisting of several hundred variables. We use 256 dimensions in the experiments for consistency. However, more research is required to understand $\alpha$-MDF's performance when filtering over thousands of variables. As with any deep learning approaches, hyper-parameter tuning may be required to produce high-performing models. Another practical observation is that utilizing more modalities does not always translate to improved performance, which is consistent with findings in [47]. Including redundant modalities can impose longer training times and pose greater difficulty for the model in extracting valuable information from the input modalities. A pre-processing step for feature selection may be advisable.

**Acknowledgments**

This work has received partial support from the National Science Foundation under grants CNS-1932068, IIS-1749783. Additionally, partial support has been provided by JSPS KAKENHI Grant Numbers 22H03671 and 22K19815. We would like to sincerely acknowledge the valuable comments and feedback provided by the reviewers. Our gratitude also goes to Yuhei Yoshimitsu for assisting in the data collection with the tensegrity robot. Furthermore, we would like to express our appreciation for the insightful discussions and constructive feedback received from Fabian Weigend and Shubham Sonawani during the review process.

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

# A   Details in $\alpha$-MDF

This section provides a detailed overview of the previously mentioned $\alpha$-MDF modules, and describes differentiable Ensemble Kalman filters as the underlying DFs framework for $\alpha$-MDF.

## A.1   Model Initialization and Embedding Functions

An auxiliary model $\mathcal{A}$ is supplied in the filtering process to support training by starting the filter via projecting the actual state $\boldsymbol{x}_{t-N:t-1}$ from low-dimensional space to latent space. The model is implemented using stochastic neural networks (SNNs) [48],

$$\mathbf{x}_{t-N:t-1}^i \sim \mathcal{A}(\mathbf{x}_{t-N:t-1}^i | \boldsymbol{x}_{t-N:t-1}), \ \forall i \in E, \tag{7}$$

where $\mathbf{x}_{t-N:t-1}^i$ is one latent state, the latent state ensemble is obtained by sampling $\mathcal{A}$ for $E$ times. During inference, we employ the trained sensor encoders' output, which is the latent representation of RGB, depth, or proprioception, as the initial state to initiate the filtering process.

Regarding the prediction step of $\alpha$-MDF, we apply positional embedding layers (sinusoidal functions) [38] in the transformer process model (Eq. 3) to generate $\boldsymbol{e}_{t-N:t-1}$ as the embedding for time-series data, $\boldsymbol{e}_{t-N:t-1} = f_{\mathcal{L}}(\mathbf{X}_{t-N:t-1}) \in \mathbb{R}^{d_x \times (N-1)}$. The positional embedding layer is utilized to label the state by index it with time $t$. When activating the action $\mathbf{a}_t$ in the process model, we also utilize a type embedding layer that indexes $\boldsymbol{e}_{t-N:t-1}$ and $\mathbf{a}_t$ with 0 and 1, and then fed to sinusoidal functions. Subsequently, the element-wise summation of outputs obtained from the aforementioned procedures serve as input to the transformer process model for further processing.

## A.2   Differentiable Ensemble Kalman Filter

Unlike prior proposals for differentiable filters, such as dEKF [9] and DPF [28], Differentiable Ensemble Kalman Filter [6] leverages recent advancements in stochastic neural networks (SNNs) [48]. Specifically, we draw inspiration from the work in [49], which established a theoretical connection between the Dropout training algorithm and Bayesian inference in deep Gaussian processes. As a result, we can use stochastic forward passes to produce empirical samples from the predictive posterior of a neural network trained with Dropout. Hence, for the purposes of filtering, we can implicitly model the process noise by sampling state from a neural network trained on the transition dynamics, i.e., $\mathbf{x}_t \sim f_{\boldsymbol{\theta}}(\mathbf{x}_{t-1})$. In contrast to previous approaches [28, 9], the transition network $f_{\boldsymbol{\theta}}(\cdot)$ models the system dynamics, as well as the inherent noise model in a consistent fashion without imposing diagonality.

**Prediction Step**: Similar to $\alpha$-MDF, we use an initial ensemble of $E$ members to represent the initial state distribution $\mathbf{X}_0 = [\mathbf{x}_0^1, \ldots, \mathbf{x}_0^E]$, $E \in \mathbb{Z}^+$. We leverage the stochastic forward passes from a trained state transition model to update each ensemble member:

$$\mathbf{x}_{t|t-1}^i \sim f_{\boldsymbol{\theta}}(\mathbf{x}_{t|t-1}^i | \mathbf{x}_{t-1|t-1}^i), \ \forall i \in E. \tag{8}$$

Matrix $\mathbf{X}_{t|t-1} = [\mathbf{x}_{t|t-1}^1, \cdots, \mathbf{x}_{t|t-1}^E]$ holds the updated ensemble members which are propagated one step forward through the state space. Note that sampling from the transition model $f_{\boldsymbol{\theta}}(\cdot)$ (using the SNN methodology described above) implicitly introduces a process noise.

**Update step**: Given the updated ensemble members $\mathbf{X}_{t|t-1}$, a nonlinear observation model $h_{\boldsymbol{\psi}}(\cdot)$ is applied to transform the ensemble members from the state space to observation space. Following our main rationale, the observation model is realized via a neural network with weights $\boldsymbol{\psi}$. Accordingly, the update equations become:

$$\mathbf{H}_t \mathbf{A}_t = \mathbf{H}_t \mathbf{X}_t - \left[ \frac{1}{E} \sum_{i=1}^{E} h_{\boldsymbol{\psi}}(\mathbf{x}_t^i), \cdots, \frac{1}{E} \sum_{i=1}^{E} h_{\boldsymbol{\psi}}(\mathbf{x}_t^i) \right], \quad (9) \quad \tilde{\mathbf{y}}_t^i \sim s(\tilde{\mathbf{y}}_t^i | \mathbf{y}_t), \ \forall i \in E. \quad (10)$$

$\mathbf{H}_t \mathbf{X}_t$ is the predicted observation, and $\mathbf{H}_t \mathbf{A}_t$ is the sample mean of the predicted observation at $t$. Traditional Ensemble Kalman Filter treats observations as random variables. Hence, the ensemble

can incorporate a measurement perturbed by a small stochastic noise to reflect the error covariance of the best state estimate [6]. In differentiable Ensemble Kalman Filter, we incorporate a Bayesian sensor encoder $s(\cdot)$. Sensor encoder serves to learn projections from observation space to latent space as in Eq. 10, where $\mathbf{y}_t$ represents the noisy sensor observation. Sampling from sensor encoder yields latent observations $\tilde{\mathbf{Y}}_t = [\tilde{\mathbf{y}}_t^1, \cdots, \tilde{\mathbf{y}}_t^{E)}]$. The KF update step can then be continued by using the learned observation and predicted observation:

$$\mathbf{K}_t = \frac{1}{E-1}\mathbf{A}_t(\mathbf{H}_t\mathbf{A}_t)^T(\frac{1}{E-1}(\mathbf{H}_t\mathbf{A}_t)(\mathbf{H}_t\mathbf{A}_t)^T + \mathbf{R})^{-1}. \tag{11}$$

The measurement noise model $\mathbf{R}$ is implemented using a multilayer perceptron (MLP), similar to the implementation in [9]. The MLP takes a learned observation $\tilde{\mathbf{Y}}_t$ at time $t$ and produces a noise covariance matrix. The final estimate of the ensemble $\hat{\mathbf{X}}_t$ is obtained by performing the measurement update step, given by:

$$\hat{\mathbf{X}}_t = \mathbf{X}_t + \mathbf{K}_t(\tilde{\mathbf{Y}}_t - \mathbf{H}_t\mathbf{X}_t). \tag{12}$$

In inference, the ensemble mean $\bar{\mathbf{x}}_{t|t} = \frac{1}{E}\sum_{i=1}^{E}\mathbf{x}_{t|t}^i$ is used as the updated state.

### A.3 Baselines

In our study, we examine two categories of baselines: (a) DFs baselines, which consist of existing methods such as those proposed in [9, 28, 26], and (b) sensor fusion strategies, as proposed in [22].

Table 4: Dimensions pertinent to each of the robot state estimation tasks.

| Method | Visual Odometry | | UR5 Manipulation | | Soft Robot | | |
| --- | --- | --- | --- | --- | --- | --- | --- |
| | State | Observation | State | Observation | State | Observation | Action |
| dEKF [9] | 5 | 2 | 10 | 10 | 7 | 7 | 40 |
| DPF [28] | 5 | 2 | 10 | 10 | 7 | 7 | 40 |
| dPF-M-lrn [9] | 5 | 2 | 10 | 10 | 7 | 7 | 40 |
| Feature Fusion [22] | - | - | 10/13 | 10/13 | 7 | 7 | 40 |
| Unimodal [22] | - | - | 10/13 | 10/13 | 7 | 7 | 40 |
| Crossmodal [22] | - | - | 10/13 | 10/13 | 7 | 7 | 40 |
| $\alpha$-MDF | 256 | 256 | 256 | 256 | 256 | 256 | 40 |

**Dimensionality:** Table 4 presents the dimensions for the state, observations, and actions utilized for each of the tasks. To ensure consistency, we opt for a dimension of 256 for $\alpha$-MDF in all tasks, thus, enabling filtering over high-dimensional spaces. Unlike the baseline methods, which use low-dimensional state definitions, we filter over higher dimension spaces with $\alpha$-MDF.

**Differentiable Filters:** To maintain consistency in the comparison of results against the DFs baselines, we train $\alpha$-MDF with a single modality. The baselines in this category include the differentiable Extended Kalman filter (dEKF) [9], differentiable particle filter (DPF) [28], and the modified differentiable particle filter (dPF-M-lrn) [9], which uses learned process and process noise models. For dEKF, the Jacobian matrix in the prediction step can either be learned end-to-end or supplied if the motion model is known. DPF employs 100 particles for both training and testing and also incorporates an observation likelihood estimation model $l$. This module takes in an image embedding and produces a likelihood that updates each particle's weight. Unlike DPF, dPF-M-lrn implements a learnable process noise model. It also adopts a Gaussian Mixture Model for calculating the likelihood for all particles. It is worth noting that all the baseline methods perform Kalman filtering on low-dimensional actual state space, whereas $\alpha$-MDF executes the filtering process in the latent space.

**Sensor Fusion:** Regarding sensor fusion baselines, we use three strategies discussed in [22], namely, Feature Fusion, Unimodal Fusion, and Crossmodal Fusion. The Feature Fusion strategy aims to process each modality individually and subsequently merge the modalities to generate a multimodal feature set using neural networks, which is then used for state estimation. The Unimodal Fusion

treats each modality $\mathcal{N} \sim (\boldsymbol{\mu}_t^{M_1}, \boldsymbol{\Sigma}_t^{M_1})$ and $\mathcal{N} \sim (\boldsymbol{\mu}_t^{M_2}, \boldsymbol{\Sigma}_t^{M_2})$ as distributions and fuse two uni-modal distribution as one normally distributed multimodal distribution $\mathcal{N} \sim (\boldsymbol{\mu}_t, \boldsymbol{\Sigma}_t)$:

$$\boldsymbol{\mu}_t = \frac{(\boldsymbol{\Sigma}_t^{M_1})^{-1}\boldsymbol{\mu}_t^{M_1} + (\boldsymbol{\Sigma}_t^{M_2})^{-1}\boldsymbol{\mu}_t^{M_2}}{(\boldsymbol{\Sigma}_t^{M_1})^{-1} + (\boldsymbol{\Sigma}_t^{M_2})^{-1}}, \quad \boldsymbol{\Sigma}_t = ((\boldsymbol{\Sigma}_t^{M_1})^{-1} + (\boldsymbol{\Sigma}_t^{M_2})^{-1})^{-1}, \qquad (13)$$

where the associative property can be used for fusing more than two modalities. For Crossmodal Fusion, information from one modality can be used to determine the uncertainty of the other ones, two coefficients are proposed as $\boldsymbol{\beta}_t^{M_1}$ and $\boldsymbol{\beta}_t^{M_2}$, where each coefficient has the same dimension of the state, the fused distribution is:

$$\boldsymbol{\mu}_t = \frac{\boldsymbol{\beta}_t^{M_1} \circ \boldsymbol{\mu}_t^{M_1} + \boldsymbol{\beta}_t^{M_2} \circ \boldsymbol{\mu}_t^{M_2}}{\boldsymbol{\beta}_t^{M_1} + \boldsymbol{\beta}_t^{M_2}}, \quad \boldsymbol{\Sigma}_t = \frac{\boldsymbol{B}_t^{M_1} \circ \boldsymbol{\Sigma}_t^{M_1} + \boldsymbol{B}_t^{M_2} \circ \boldsymbol{\Sigma}_t^{M_2}}{\boldsymbol{B}_t^{M_1} + \boldsymbol{B}_t^{M_2}}, \qquad (14)$$

where $\boldsymbol{B}_t^{M} = (\boldsymbol{\beta}_t^{M})^T \boldsymbol{\beta}_t^{M}$. As mentioned in [22], each sensor encoder was independently trained and subsequently used for end-to-end training with DFs. We adopt a similar approach, but with a differentiable Ensemble Kalman Filter backbone in place instead. The resampling procedure from the fused distribution in this scenario is achieved by using the reparematerization trick [50].

## B  Additional Experiments

This section presents supplementary experimental results for each task. For (1) Visual Odometry Tasks, we offer full detailed experiments; however, for (2) Multimodal Manipulation Tasks and (3) Soft Robot Modeling Tasks, we concentrate mainly on ablation studies.

### B.1  Visual Odometry Tasks

In this experiment, we investigate the performance of $\alpha$-MDF on the popular KITTI Visual Odometry dataset [40]. We only consider RBG images as the input modality in order to make a fair comparison with the baselines [9, 28, 26]. Following the same evaluation procedure as our baselines, we define the actual state of the moving vehicle as a 5-dimensional vector $\boldsymbol{x} = [x, y, \theta, v, \dot{\theta}]^T$, including the position and orientation of the vehicle, and the linear and angular velocity w.r.t. the current heading direction $\theta$. The raw observation $\mathbf{y}$ corresponds to the RGB camera image of the current frame and a difference image between the current frame and the previous frame, where $\mathbf{y} \in \mathbb{R}^{150 \times 50 \times 6}$ as shown in Fig. 8. The learned observation $\tilde{\mathbf{y}}$ is defined as $\tilde{\mathbf{y}} = [v, \dot{\theta}]^T$, since only the relative changes of position and orientation can be captured between two frames. We use the latent state $\mathbf{x} \in \mathbb{R}^{256}$ for $\alpha$-MDF.

**Data:** The KITTI Visual Odometry dataset includes 11 trajectories capturing the ground truth pose (translation and rotation matrices) of a vehicle navigating urban areas at a data collection rate of approximately 10Hz. To facilitate the learning process, we standardize the data by normalizing each dimension to have a mean of 0 and a standard deviation of 1 during training. To process the provided pose data, we convert

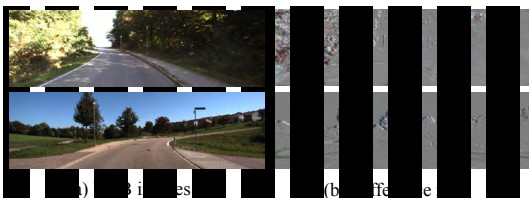

Figure 8: KITTI visual inputs.

them to quaternions to capture the minimal changes between consecutive quaternion pairs. Subsequently, the results are converted back to radians to represent the angular velocity $\dot{\theta}$. This conversion ensures that the angular velocity remains minimal and falls within the range of $[-\pi, \pi]$.

#### B.1.1  Results

The performance of state estimation is evaluated using an 11-fold cross-validation, whereby 1 trajectory is withheld at each time. The standard KITTI benchmark metrics, namely the translational error (m/m) and rotational error (deg/m), are reported in Table 5. The error metrics are computed from the test trajectory over all subsequences of 100 timesteps, as well as all subsequences of 100,

Table 5: Result evaluations on KITTI Visual Odometry task measured in m/m and deg/m denote the translational error and the rotational error.

| Method | Test 100 | | Test 100/200/400/800 | |
|---|---|---|---|---|
| | m/m | deg/m | m/m | deg/m |
| dEKF [9] | 0.2646±0.004 | 0.1386±0.002 | 0.3159±0.002 | 0.0923±0.005 |
| DPF [28] | 0.1344±0.002 | 0.1203±0.007 | 0.2255±0.001 | 0.0716±0.004 |
| dPF-M-lrn [9] | 0.1720±0.010 | 0.0974±0.009 | 0.1848±0.004 | 0.0611±0.003 |
| $\alpha$-MDF | **0.0718±0.001** | **0.0954±0.001** | **0.0379±0.002** | **0.0328±0.001** |

Means±standard errors.

200, 400, and 800 timesteps. Figure 9 presents the performance of $\alpha$-MDF and other differentiable filtering techniques. It is important to note that incorporating domain- and data-specific information, such as using stereo images [51], integrating LiDAR [52, 53], or applying SLAM and loop-closure related assumptions [51, 54], can yield lower error metrics. However, to ensure fair and comparable evaluations, we utilize the most commonly used setup when comparing filtering techniques in a task-agnostic fashion (as performed in [9, 28, 26]).

Table 5 presents the outcomes of our proposed method in comparison with the existing state-of-the-art DFs, namely dEKF, DPF, and dPF-M-lrn. In order to provide a fair comparison, we do not include unstructured LSTM models as baselines since prior works [26, 9] have shown that they do not achieve comparable results. The pre-trained sensor encoder with the same visual inputs is used and integrated into all the DF frameworks evaluated. In this experiment, the motion model of the vehicle is known, and the only unknown

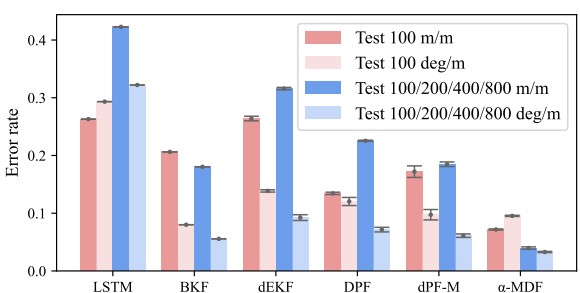

Figure 9: Visual Odometry results with different differentiable filters: the error rate for LSTM and BKF are reported from [26], dEKF, DPF, and dPF-M are reproduced.

part of the state is the velocities. In light of the above, we adopt a learnable process model to update state variables alongside an established motion model to update the $(x, y, \theta)$ variables. While the computed Jacobian matrix is supplied in training and testing for dEKF, our $\alpha$-MDF demonstrates significant improvements compared to dEKF, DPF, and dPF-M-lrn. Specifically, we observed a reduction in the translational error of approximately 88%, 83%, and 79% for Test 100/200/400/800. The results also reflect a considerable reduction in rotational error of approximately 64%, 54%, and 46% as compared to each of the baselines. Our analysis of $\alpha$-MDF reveals that conducting filtering on high-dimensional observations in the latent space yields better results than conducting filtering on the actual state space.

### B.1.2 Compare to EKF

In this section, we present a comparison of the results obtained from a non-learning Extended Kalman Filter (EKF) and $\alpha$-MDF on the KITTI Visual Odometry task. As previously mentioned, the actual state of the moving vehicle is represented by a 5-dimensional vector $\boldsymbol{x} = [x, y, \theta, v, \dot{\theta}]^T$, while the observation $\tilde{\mathbf{y}}$ is defined as $\tilde{\mathbf{y}} = [v, \dot{\theta}]^T$. The EKF can be formulated using the provided analytical model.

$$\boldsymbol{x}_t = f(\boldsymbol{x}_{t-1}) = \mathbf{A}\boldsymbol{x}_{t-1} + \mathbf{q}_t \quad \mathbf{q}_t \sim \mathcal{N}(0, \mathbf{Q}_t),$$
$$\tilde{\mathbf{y}} = h(\boldsymbol{x}_t) + \mathbf{r}_t = \mathbf{H}\boldsymbol{x}_t + \mathbf{r}_t \quad \mathbf{r}_t \sim \mathcal{N}(0, \mathbf{R}_t). \tag{15}$$

where $\mathbf{H}$ is identity matrix $\mathbf{H} = \left[\begin{smallmatrix} 0 & 0 & 0 & 1 & 0 \\ 0 & 0 & 0 & 0 & 1 \end{smallmatrix}\right]$. The EKF prediction step is:

$$\hat{\boldsymbol{x}}_t = \mathbf{A}\boldsymbol{x}_{t-1} + \mathbf{q}_t, \quad \hat{\boldsymbol{\Sigma}}_t = \mathbf{F}\boldsymbol{\Sigma}_{t-1}\mathbf{F}^T + \mathbf{Q}_t. \tag{16}$$

where the Jacobian of the process model can be supplied via Taylor expansion,

$$A = \begin{bmatrix} 1 & 0 & 0 & \sin\theta\Delta t & 0 \\ 0 & 1 & 0 & \cos\theta\Delta t & 0 \\ 0 & 0 & 1 & 0 & \Delta t \\ 0 & 0 & 0 & 1 & 0 \\ 0 & 0 & 0 & 0 & 1 \end{bmatrix}, \quad \mathbf{F} = \frac{\partial f(\boldsymbol{x}_{t-1})}{\partial \boldsymbol{x}_{t-1}} = \begin{bmatrix} 1 & 0 & v\cos\theta\Delta t & \sin\theta\Delta t & 0 \\ 0 & 1 & -v\sin\theta\Delta t & \cos\theta\Delta t & 0 \\ 0 & 0 & 1 & 0 & \Delta t \\ 0 & 0 & 0 & 1 & 0 \\ 0 & 0 & 0 & 0 & 1 \end{bmatrix}. \tag{17}$$

The update step for EKF is:

$$\mathbf{S}_t = \mathbf{H}\hat{\boldsymbol{\Sigma}}_t\mathbf{H}^T + \mathbf{R}_t, \quad \mathbf{K}_t = \hat{\boldsymbol{\Sigma}}_t\mathbf{H}^T\mathbf{S}_t^{-1},$$
$$\boldsymbol{x}_t = \hat{\boldsymbol{x}}_t + \mathbf{K}_t(\tilde{\mathbf{y}} - \mathbf{H}\hat{\boldsymbol{x}}_t), \quad \boldsymbol{\Sigma}_t = (\mathbf{I} - \mathbf{K}_t\mathbf{H})\hat{\boldsymbol{\Sigma}}_t. \tag{18}$$

To ensure a fair and unbiased comparison, both the Extended Kalman Filter (EKF) and $\alpha$-MDF models are provided with the same low-dimensional observation $\tilde{\mathbf{y}} + \epsilon$, where $\epsilon$ is a noise sample obtained from a Gaussian distribution $\mathcal{N} \sim (0, \begin{bmatrix} 1.5 & 0 \\ 0 & 0.1 \end{bmatrix})$.

We report a comparison via translational and rotational errors in Table 6. For the EKF model, the noise covariance matrices $\mathbf{Q}_t$ and $\mathbf{R}_t$ are manually fine-tuned. Additionally, we initialize the filter with $\boldsymbol{\Sigma}_0 = \mathbf{I}$. For $\alpha$-MDF, we keep the same framework as

Table 6: Comparison between EKF and $\alpha$-MDF.

| Method | Test 100 | | Test 100/200/400/800 | |
|---|---|---|---|---|
| | m/m | deg/m | m/m | deg/m |
| EKF | 0.2391±0.02 | 0.1548±0.02 | 0.2757±0.03 | 0.0623±0.01 |
| $\alpha$-MDF | 0.1642±0.02 | 0.0593±0.01 | 0.1509±0.01 | 0.0327±0.01 |
| Means±standard errors. | | | | |

when filtering over a latent state with 256 dimensions. However, we substitute the sensor encoder from $s^1$ to $s^2$ (refer to Table 12). This modification allows for projecting the low-dimensional observation into the latent space. Our observations indicate that $\alpha$-MDF, when utilizing attention gain, reduces the error over the EKF. $\alpha$-MDF has the additional benefit of automatically learning noise profiles during the training process, thereby eliminating the manual tuning step required by the EKF.

### B.1.3  6D Motion State

To conduct a more comprehensive investigation into the visual odometry task, we extended our analysis by employing a larger state space. In this section, we consider the 6D motion of the vehicle where 3 different heading directions are defined namely yaw $\theta$, pitch $\psi$, and roll $\phi$. The actual state is defined as $\boldsymbol{x} = [x, y, z, \phi, \psi, \theta, v_1, v_2, v_3, \dot{\phi}, \dot{\psi}, \dot{\theta}]^T$. Similar to the previous setup, $\alpha$-MDF takes the image pair at $t-1$ and $t$ as input, with the observation $\tilde{\mathbf{y}}$ defined as $\tilde{\mathbf{y}} = [v_1, v_2, v_3, \dot{\phi}, \dot{\psi}, \dot{\theta}]^T$.

The comparison results for translational and rotational errors, with an increased state space, are presented in Table 7. Notably, the performance of $\alpha$-MDF remains stable along the yaw axis, as observed in comparison to

Table 7: Using 6D motion state for $\alpha$-MDF.

| Method | Axis | Test 100 | | Test 100/200/400/800 | |
|---|---|---|---|---|---|
| | | m/m | deg/m | m/m | deg/m |
| $\alpha$-MDF | Yaw $\theta$ | 0.072±0.001 | 0.097±0.004 | 0.041±0.003 | 0.033±0.001 |
| $\alpha$-MDF | Pitch $\psi$ | 0.032±0.002 | 0.013±0.001 | 0.028±0.003 | 0.019±0.001 |
| $\alpha$-MDF | Roll $\phi$ | 0.033±0.004 | 0.032±0.010 | 0.049±0.001 | 0.029±0.002 |
| Means±standard errors. | | | | | |

the results reported in Table 5. Additionally, we observe smaller translational and rotational errors on the pitch and roll axes. This observation can be attributed to the relatively minor deviations on the $z$ axis, even during inclined maneuvers such as ascending or descending a hill. In conclusion, when the search space expands to include a larger state space, $\alpha$-MDF demonstrates comparable results, indicating its ability to handle increased complexity and maintain performance.

### B.2  Multimodal Manipulation Tasks

**Task Setup:** For $\alpha$-MDF, we define the latent state $\mathbf{x} \in \mathbb{R}^{256}$ for all the manipulation tasks. The actual state of the UR5 robot is described by $\boldsymbol{x}_R$, which consists of the seven joint angles $(J_1\text{-}J_7)$

Table 8: Ablation study on UR5 manipulation task with different combination of the modalities.

| | RGB | Depth | Joint | F/T | Joint (deg) | EE (cm) | Obj (cm) |
|---|---|---|---|---|---|---|---|
| Task (1) | ✓ | | | | 2.78±0.09 | 1.06±0.01 | - |
| | | ✓ | | | 3.65±0.10 | 1.38±0.05 | - |
| | | | ✓ | | 9.53±0.20 | 3.22±0.14 | - |
| | | ✓ | ✓ | | 2.39±0.11 | 1.01±0.02 | - |
| | ✓ | | ✓ | | 2.69±0.01 | 1.09±0.03 | - |
| | ✓ | ✓ | | | **1.91±0.08** | **0.64±0.03** | - |
| | ✓ | ✓ | ✓ | | 2.19±0.09 | 0.75±0.01 | - |
| Task (2) | ✓ | | | | 7.49±0.06 | 3.81±0.17 | - |
| | | | ✓ | | 5.47±0.08 | 3.32±0.04 | - |
| | ✓ | | ✓ | | **5.24±0.04** | **3.04±0.01** | - |
| Task (3) | | | ✓ | ✓ | 2.93±0.01 | 2.26±0.02 | 3.26±0.01 |
| | ✓ | | ✓ | ✓ | 3.16±0.20 | 2.34±0.04 | 3.66±0.30 |
| | ✓ | ✓ | | ✓ | 1.42±0.08 | 0.93±0.01 | **1.47±0.02** |
| | ✓ | ✓ | ✓ | | **1.37±0.02** | 0.94±0.01 | 1.78±0.06 |
| | ✓ | ✓ | ✓ | ✓ | 1.41±0.04 | **0.90±0.01** | 1.65±0.01 |

Means±standard errors.

and the Cartesian coordinates $(x, y, z)$ of the robot's end-effector. This Cartesian coordinate system is centered at the manipulation platform's origin point $(0, 0, 0)$. On the other hand, the state of the object being manipulated is represented by $\boldsymbol{x}_O$, which only includes the Cartesian coordinates $(x, y, z)$ of the object. The input modalities for each of the three tasks differ. In task (1), input is given through three modalities: $\mathbf{y}^1$, $\mathbf{y}^2$, and $\mathbf{y}^3$. The first modality $\mathbf{y}^1 \in \mathbb{R}^{224 \times 224 \times 3}$ is a camera image captured from a frontal angle. The second modality $\mathbf{y}^2 \in \mathbb{R}^{224 \times 224 \times 1}$ depicts depth maps from the same camera view. Lastly, $\mathbf{y}^3$ is a proprioceptive input source with dimensions $\mathbb{R}^7$, representing the joint angles' values. In this task, the proprioceptive input specifically refers to the joint angles as the source. In task (2), input is given by only two modalities: $\mathbf{y}^1$ and $\mathbf{y}^3$, but from a real-world perspective. In task (3), input is received from four modalities: $\mathbf{y}^1$, $\mathbf{y}^2$, $\mathbf{y}^3$, and $\mathbf{y}^4$. $\mathbf{y}^4$ contains the Force/torque (F/T) sensor readings from the robot gripper, where $\mathbf{y}^4 \in \mathbb{R}^6$, while the first two modalities are identical to task (1).

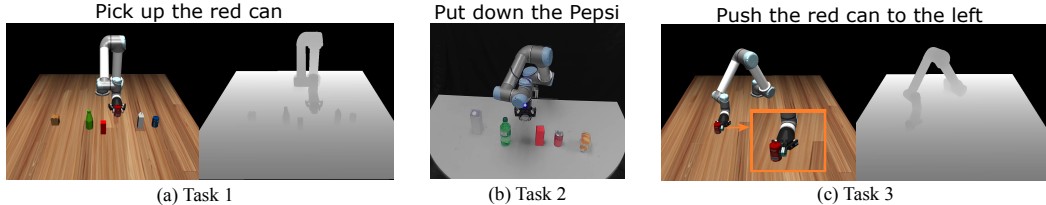

| Pick up the red can | Put down the Pepsi | Push the red can to the left |
|---|---|---|
| (a) Task 1 | (b) Task 2 | (c) Task 3 |

Figure 10: The multimodal manipulation experiment involves the following subtasks: (a) Task 1 utilizing RGB, depth, and joint modalities, (b) Task 2 utilizing only RGB and joint modality, and (c) Task 3 utilizing RGB, depth, joint, and Force/torque (F/T) sensor modalities. The F/T sensor is mounted on the grabber, as depicted by the orange box.

**Data:** Data collection is conducted for both simulation with MuJoCo [55] and real-world scenarios. We record the UR5 robot operating on a random object by performing one of "pick", "push", and "put down" actions. We collect 2,000 demonstrations in simulation for task (1), and 100 on the real robot for task (2), with changing the location of each object for each demonstration. For task (3), we collect 2,000 demonstrations in simulation with adding the tactile sensors. We use ABR control and robosuite [56] in addition to MuJoCo to ensure rigorous dynamics in the simulator. Each demonstration sequence has a length of approximately 350 steps with a timestep of 0.08 seconds. An 80/20 data split is utilized for training and testing each task. It should be noted that in all tasks, we normalize the joint modality $\mathbf{y}^3$ and apply Gaussian noise to each joint angle, drawn from the distribution $\mathcal{N} \sim (0, \sigma^2 \mathbf{I})$ where $\sigma^2 = 0.1$. We collect the F/T sensor readings directly from MuJoCo's native touch sensor. Moreover, the depth maps obtained from MuJoCo are with no noise therefore can be regarded as high-fidelity data.

### B.2.1 Ablation Study

In addition to the findings presented in Section 4.2, we perform a comprehensive ablation analysis for each manipulation task to address the question, "How does the use of multiple modalities compare to a subset of modalities for state estimation with differentiable filters?". Table 8 displays the outcome for each task with various number of modalities using MAE metric. The highest margin of error is indicated by the red shading, while the complete modality is labeled by green shading for each task. Interestingly, even though using all modalities can generate comparable results, in certain tasks, utilizing all modalities does not necessarily guarantee superior performance compared to utilizing a subset of modalities. Through our experiments in Task (1), it becomes apparent that the optimal performance is achieved by utilizing the subset of modalities $[\mathbf{y}^1, \mathbf{y}^2]$, which yields an improvement of joint angles ($2.19° \rightarrow 1.91°$). In Task (3), we observe that diverse subsets of modalities lead to superior state estimation results for joint angles, EE, and the object locations respectively. Analysis of Table 8 indicates an important role played by the depth map $\mathbf{y}^2$ when considering all observations. This suggests that $\mathbf{y}^2$ is treated as high-fidelity data during training, thereby contributing the most towards the final results.

Henceforth, we conduct an additional ablation analysis to ascertain whether or not the use of a combination of high-fidelity and low-fidelity sensor inputs offers a potential benefit. As noted during data collection, the proprioceptive input $\mathbf{y}^3$ comprising joint angles is obtained via adding Gaussian noise and is therefore considered a low-fidelity input. Figure 11 illustrates the scenario of using $\mathbf{y}^3$ and not using $\mathbf{y}^3$ while applying distinct levels of Gaussian blur in the image and depth space. Notably, without employing $\mathbf{y}^3$, the state estimation per-

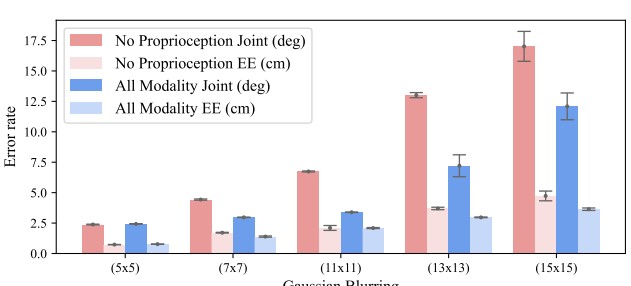

Figure 11: State estimation results are shown after introducing diverse levels of noise to $[\mathbf{y}^1, \mathbf{y}^2]$. The red group depicts results using $[\mathbf{y}^1, \mathbf{y}^2]$ modality, while the blue group represents results using $[\mathbf{y}^1, \mathbf{y}^2, \mathbf{y}^3]$ modality.

formance deteriorates as the level of blur increases. On the other hand, $\mathbf{y}^3$ - despite being classified as a low-fidelity modality - contributes to the final state estimation. In particular, at the highest level of blur, incorporating $\mathbf{y}^3$ yields a 29% improvement in joint angle estimation and a 17% improvement for end-effector locations.

### B.2.2 Sensitivity Analysis

In this study, we analyze the effects of three key factors on the performance of $\alpha$-MDF. These factors are latent dimensions, the length of previous states, the number of latent ensemble members, whether using Transformers or Multilayer Perceptrons (MLPs) as process model, and with or without the matrix $\tilde{\boldsymbol{M}}$. Our investigation focuses on understanding how these factors impact the overall performance of the $\alpha$-MDF framework. In this experiment, we use robot manipulation task (1) as an example.

The findings from the sensitivity analysis are summarized in Table 9. Regarding latent dimensions, we observe that a larger latent dimension does not consistently yield better error metrics. The optimal latent dimension may vary for different tasks. Regarding the length of the previous state, we find that using $\mathbf{X}_{t-10:t-1}$ leads to more accurate results compared to using $\mathbf{X}_{t-30:t-1}$. This suggests that a longer history of states may not significantly contribute to estimating the current state. Therefore, we recommend using a smaller or medium window size for state transition models. As for the number of ensemble members, using a larger value for $E$ does improve accuracy. However, it is worth noting that increasing the number of ensemble members can result in a larger state space, which may introduce inefficiency. In terms of utilizing Transformer-style neural networks for process models, the results from Table 9 indicate an advantage for this approach, as indicated

Table 9: Sensitivity analysis within the $\alpha$-MDF framework, focusing on three factors: latent dimensions, length of previous states, and the number of ensemble members.

| | RGB | Depth | Joint | F/T | Joint (deg) | EE (cm) |
|---|---|---|---|---|---|---|
| $\alpha$-MDF with 64 latents | ✓ | ✓ | ✓ | | 2.54±0.06 | 0.87±0.04 |
| $\alpha$-MDF with 256 latents | ✓ | ✓ | ✓ | | 2.19±0.09 | 0.75±0.01 |
| $\alpha$-MDF with 512 latents | ✓ | ✓ | ✓ | | 2.38±0.01 | 0.82±0.01 |
| $\alpha$-MDF with $\mathbf{X}_{t-5:t-1}$ | ✓ | ✓ | ✓ | | 2.19±0.09 | 0.75±0.01 |
| $\alpha$-MDF with $\mathbf{X}_{t-10:t-1}$ | ✓ | ✓ | ✓ | | 2.16±0.06 | 0.83±0.04 |
| $\alpha$-MDF with $\mathbf{X}_{t-30:t-1}$ | ✓ | ✓ | ✓ | | 2.72±0.03 | 0.79±0.07 |
| $\alpha$-MDF with $E = 2^3$ | ✓ | ✓ | ✓ | | 2.67±0.12 | 1.10±0.02 |
| $\alpha$-MDF with $E = 2^5$ | ✓ | ✓ | ✓ | | 2.19±0.09 | 0.75±0.01 |
| $\alpha$-MDF with $E = 2^7$ | ✓ | ✓ | ✓ | | 1.77±0.05 | 0.67±0.01 |
| $\alpha$-MDF with MLPs | ✓ | ✓ | ✓ | | 2.45±0.05 | 0.83±0.02 |
| $\alpha$-MDF with no $\tilde{\boldsymbol{M}}$ | ✓ | ✓ | ✓ | | 7.25±0.05 | 2.23±0.09 |

Means±standard errors.

by the green-shaded row. However, it is important to acknowledge that for certain non-complex tasks, employing lightweight MLPs as process models can also be a suitable option. It is crucial to consider the specific task requirements and complexity when deciding between Transformer-style neural networks and MLPs as process models.

### B.2.3   Mask in Attention Gain

Within the attention gain module, we incorporate a matrix $\tilde{\boldsymbol{M}}$ that selectively preserves diagonal elements of the attention map. This approach is based on the assumption that within each latent vector, each index possesses probabilistic independence. To empirically verify this assumption, we conducted an additional experiment where we trained an alternative $\alpha$-MDF framework. In this framework, we deliberately excluded the matrix $\tilde{\boldsymbol{M}}$ in the attention gain module for the specific purpose of evaluating the effect on robot manipulation task (1). The results of our experiment are reported in Table 9 as indicated by the red-shaded row. It shows a significant increase in the Mean Absolute Error (MAE) for joint angle estimation when the causality-enforced map $\tilde{\boldsymbol{M}}$ was excluded from the attention gain module. Specifically, the MAE increased from $2.19°$ to $7.25°$. Moreover, the MAE for tracking the end effector also deteriorated from 0.75cm to 2.23cm. Based on these results, it is strongly recommended to utilize the causality-enforced map $\tilde{\boldsymbol{M}}$ within the attention gain module for improved performance in both joint angle estimation and end effector tracking.

### B.3   Soft Robot Modeling Tasks

This section presents a comprehensive analysis of the tensegrity robot structure, the bending motion mechanism, and pertinent sensory information, followed by a description of additional experimental outcomes related to this task.

**Preliminaries**: Our research utilizes a tensegrity robot arm (developed in [45]) that follows a strict tensegrity structure featuring struts, cables (including spring-loaded and actuated cables), and five layers of arm-like tensegrity structures, which

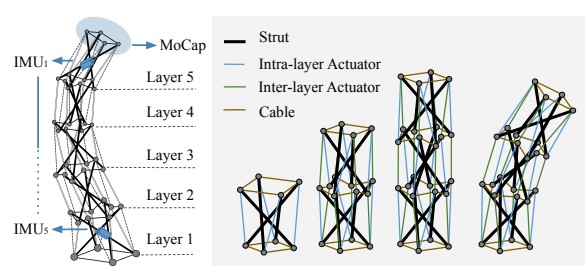

Figure 12: The tensegrity robot features 5 flexible layers, each a tensegrity module with struts, cables, and actuators.

produce continuous bending postures when exposed to external forces. The longitudinal length is maintained by stiff cables, while the bending direction is solely determined by external forces. We determine the robot's kinematics through data from Inertial Measurement Units (IMUs), optical motion capture (MoCap), and proportional pressure control valves, with each of the five struts in each

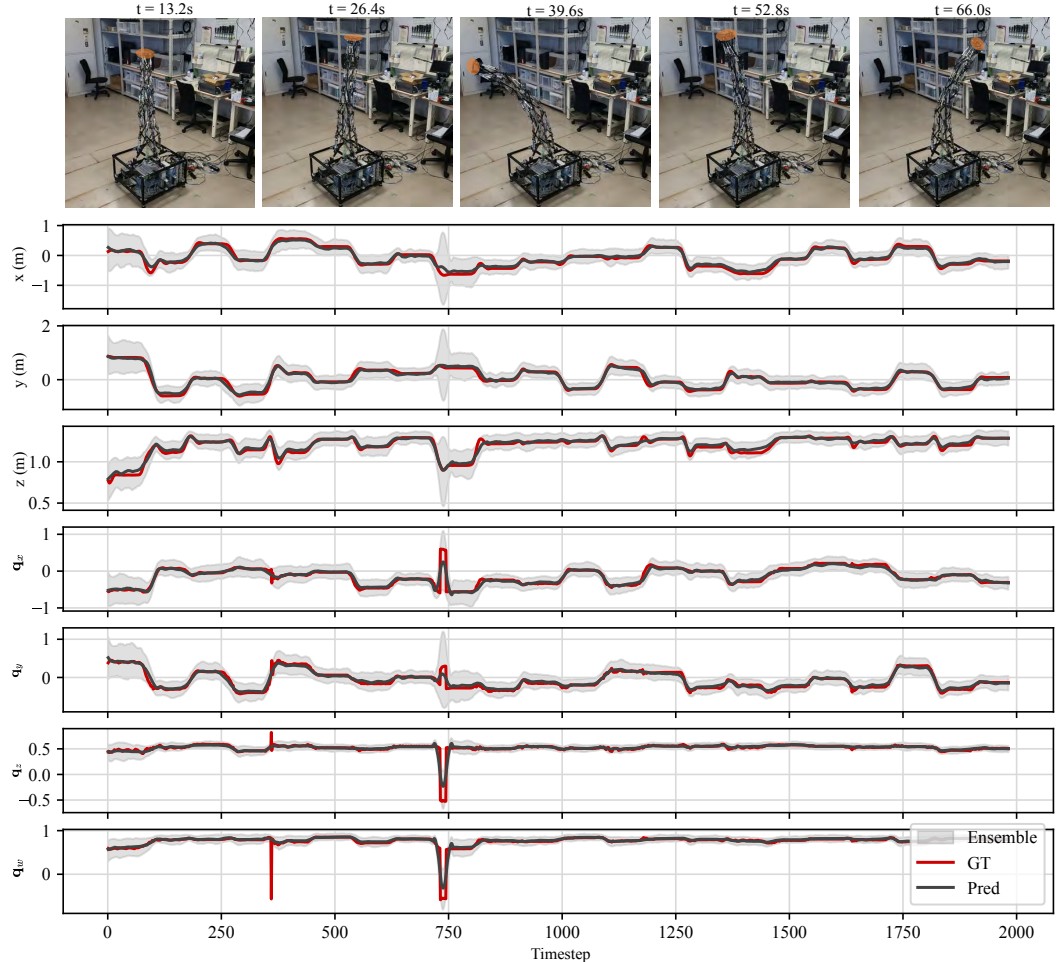

Figure 13: Predicted end-effector (EE) positions and quaternion vectors **q** in the soft robot modeling task. The **top** row displays the actual robot posture at the corresponding time, with the orange circle indicating the EE positions, which are not included in the RGB modality input.

layer featuring an IMU. We also record the video by placing a camera in front of the robot while collecting all sensory data.

A soft robot's state at $t$ is a 7-dimensional vector $\mathbf{x}_t = [x, y, z, \mathbf{q}_x, \mathbf{q}_y, \mathbf{q}_z, \mathbf{q}_w]^T$, indicating its position and orientation relative to the base frame (layer 1's bottom). $\mathbf{q}$ represents the robot's posture. The system's action is the pressure vector of its 40 pneumatic cylinder actuators ($\mathbf{a}_t \in \mathbb{R}^{40}$). Its raw observation is comprised of 5 IMU readings ($\mathbf{y}_t^3 \in \mathbb{R}^{30}$), with each IMU measuring a 6-dimensional vector of accelerations and angular velocities relative to its location. Fig. 12 illustrates the locations of the IMUs on the struts (blue cubes) in each layer.

**Data**: The complete set of modalities comprises $[\mathbf{y}^1, \mathbf{y}^2, \mathbf{y}^3]$, where $\mathbf{y}^1 \in \mathbb{R}^{224 \times 224 \times 3}$ represents RGB images, $\mathbf{y}^2 \in \mathbb{R}^{224 \times 224}$ is synthetic depth maps which we generate from DPT repo [46] utilizing "Intel/dpt-large", and $\mathbf{y}^3 \in \mathbb{R}^{30}$ is proprioceptive inputs (IMUs). The dataset is generated by performing optical motion capture on the real tensegrity robot hand tip while randomly supplying desired pressure vectors to the pneumatic cylinder actuators. The action $\mathbf{a}_t \in \mathbb{R}^{40}$, 5 IMU readings $\mathbf{y}_t^3 \in \mathbb{R}^{30}$, and a 7-dimensional state $\mathbf{x}_t$ are recorded, with 40-dimensional pressure vectors being used as a control signal. A total of 12,000 trials of robot motion are collected, with each trial involving moving the robot from its current equilibrium posture to the next equilibrium posture by applying the new desired pressure. All data are collected via a ROS2 network with a sampling frequency of 30Hz and are synchronized using the "message_filters" package.

### B.3.1 Ablation Study

In addition to the results presented in Section 4.3, we evaluate various combinations of modalities to determine whether an optimal subset of modalities can be identified to attain comparable outcomes without using all modalities during the filtering operation. As demonstrated in Table 10, utilizing only one modality fails to achieve comparable results, with the highest accuracy (2.07cm) exclusively from employing $\mathbf{y}^1$ (RGB). The lowest error in pos-

Table 10: Ablation study on Tensegrity robot.

| RGB | Depth | IMUs | EE (cm) | $\mathbf{q}(10^1)$ |
|:---:|:---:|:---:|:---:|:---:|
| ✓ | | | 2.07±0.03 | 0.31±0.08 |
| | ✓ | | 2.77±0.01 | 0.19±0.05 |
| | | ✓ | 8.99±0.02 | 0.79±0.03 |
| | ✓ | ✓ | 2.08±0.03 | 0.14±0.02 |
| ✓ | | ✓ | 1.73±0.05 | 0.12±0.02 |
| ✓ | ✓ | | 1.74±0.06 | **0.10±0.02** |
| ✓ | ✓ | ✓ | **1.67±0.09** | 0.12±0.01 |

Means±standard errors.

ture estimation for the robot is obtained by leveraging $[\mathbf{y}^1, \mathbf{y}^2]$, showing slight improvement (0.10→0.12) over leveraging the full modalities $[\mathbf{y}^1, \mathbf{y}^2, \mathbf{y}^3]$. However, the lowest MAE error for the EE position persists even when all modalities are employed. Interestingly, using solely $\mathbf{y}^3$ results in the highest state estimation error, which aligns with the lowest attention value visualized in Fig 14. As depicted in Fig. 14, it is evident that $\alpha$-MDF prioritizes $\mathbf{y}^1$ over other modalities. Interestingly, the attention values change upon turning off certain modalities while the system remains stable and functional.

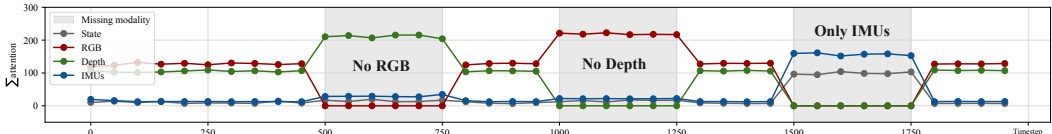

Figure 14: The corresponding accumulated attention values for each modality during testing. The gray areas show certain modalities are selected or not selected.

### B.3.2 Concept Drifts

To investigate the effects of concept drift and contextual changes [57] on the $\alpha$-MDF framework, we incorporated a background change at inference time. In particular, image blending is used to overlay a different RGB picture into the background. The objective of the experiment is to inference behavior when changes to the environment occur. We evaluated the tracking performance at various blending levels, as illustrated in Fig. 15. The results provide an understanding of how effectively the $\alpha$-MDF framework handles concept drift at different levels of intensity. It is noteworthy that despite substantially affecting the visual representation of the scene the achieved results (6.54cm) are comparable to utilizing only IMUs (8.99cm). This experiment provides an early insight into the utility of multimodality for mitigating the adverse effects due to contextual changes and concept drift.

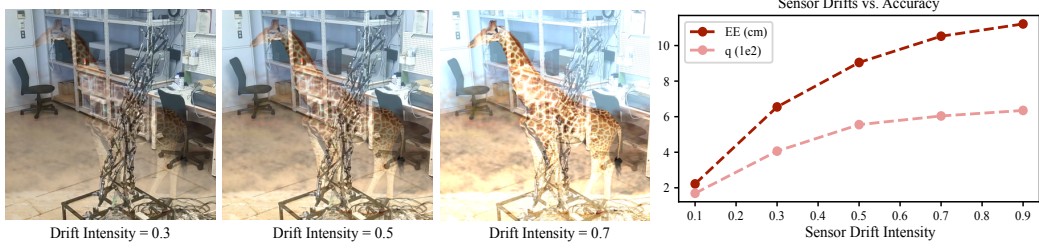

Figure 15: Concept drifts analysis by adding background change with scale in RGB space.

# C   Complexity and Training Details

In this section, we present an analysis of the computational complexity associated with each task by measuring the wall-clock time. Additionally, we provide comprehensive information regarding the model hyper-parameters and training curriculum employed for the experiments. These details offer insights into the computational requirements and settings utilized for training the models in our study.

## C.1   Complexity

To assess the computational complexity of the proposed $\alpha$-MDF framework alongside the baseline differentiable filters (DFs), we measured the wall-clock time during inference. The results, provided in Table 11, demonstrate the computational time for each approach. In the comparison with DFs baselines, we only considered a single modality. It is worth noting that in the multimodality setting, we observed only a marginal increase in the elapsed time (0.03 sec) when handling multiple types of observations. This indicates that the proposed framework, $\alpha$-MDF, is efficient and capable of effectively handling various modalities without significantly compromising computational performance.

Table 11: Wall-clock time (sec) for each task.

|  | Modality | Visual Odometry task | Robot Manipulation | | | Soft Robot task |
|---|---|---|---|---|---|---|
|  |  |  | task(1) | task(2) | task(3) |  |
| dEKF [9] | 1 | 0.0463±0.004 | 0.0469±0.003 | 0.0472±0.002 | - | 0.0474±0.003 |
| DPF [28] | 1 | 0.0486±0.005 | 0.0515±0.002 | 0.0509±0.002 | - | 0.0600±0.004 |
| dPF-M-lrn [9] | 1 | 0.0693±0.011 | 0.0854±0.001 | 0.0844±0.002 | - | 0.0590±0.002 |
| $\alpha$-MDF | 1 | 0.0547±0.002 | 0.0554±0.011 | 0.0524±0.003 | - | 0.0633±0.003 |
| $\alpha$-MDF | ≥2 | - | 0.0836±0.002 | 0.0873±0.004 | 0.0890±0.005 | 0.0910±0.004 |

Means±standard errors.

## C.2   Training Details

Table 12 provides an exhaustive enumeration of all learnable modules utilized in $\alpha$-MDF, which includes three primary components: the state transition model $f_{\boldsymbol{\theta}}$, the sensor encoders $[s^1(\cdot), s^2(\cdot), \cdots, s^M(\cdot)]$, and the attention gain (AG) module. We adopt self-attention layers with dimension 256 and 8 heads, denoted as "Self Attn", in the state transition model. The cross-attention layers, denoted as "Cross Attn", is with dimension 32 and 4 heads in the AG module. The sensor encoders utilized in our approach and all baseline models are identical, with $s^1$ acting on image-like modalities, utilizing ResNet18 [58] for learning high-dimensional observation representations, while $s^2$ pertains to low-dimensional modalities such as joint angles. The auxiliary model $\mathcal{A}$ and the decoder $\mathcal{D}$ shares a similar structure to $s^2$, but with different number of neurons. Note that $x$ is the dimension of the actual state.

Table 12: $\alpha$-MDF's learnable sub-modules.

| | |
|---|---|
| $f_{\boldsymbol{\theta}}$: | $3\times$ SNN(256, ReLu), Positional Embedding, $3\times$ Self Attn(256,8), $2\times$ SNN(256, ReLu), $1\times$ SNN($d_x$, -) |
| $s^1$: | $1\times$ ResNet18(h,w,ch), $2\times$ fc(2048, ReLu), $1\times$ SNN(512, ReLu), $1\times$ SNN($d_x$, -) |
| $s^2$: | $1\times$ SNN(128, ReLu), $1\times$ SNN(256, ReLu), $1\times$ SNN(512, ReLu), $1\times$ SNN($d_x$, -) |
| AG: | Positional Embedding, $1\times$ Cross Attn(32, 4, mask) |
| $\mathcal{A}$: | $1\times$ SNN(128, ReLu), $1\times$ SNN(256, ReLu), $1\times$ SNN(512, ReLu), $1\times$ SNN(1024, ReLu), $1\times$ SNN($d_x$, -) |
| $\mathcal{D}$: | $1\times$ fc(256, ReLu), $1\times$ SNN(128, ReLu), $1\times$ SNN(32, ReLu), $1\times$ SNN($x$, -) |

fc: Fully Connected, SNN: Stochastic Neural network.

During $\alpha$-MDF training, we employ the curriculum outlined in Algorithm 1. Note that some tasks may require pre-training the sensor encoders before performing end-to-end training the entire framework. For each task, we train $\alpha$-MDF model with utilizing batch size of 64 on a single NVIDIA A100 GPU for roughly 48 hours. For all the tasks, we use the Adamw [59] optimizer with a learning rate of 1e-4.

---

**Algorithm 1** Condition in Latent Space: training algorithm return the weights $\omega$

---

**Input:** $\alpha$-MDF, dataloader $\left(\{\boldsymbol{x}_t\}_{t-N}^{t+1}, \{\mathbf{y}_t^m\}_{m=1}^M, \{\mathbf{y}_{t+1}^m\}_{m=1}^M, \{\mathbf{a}_t\}_{t-1}^{t+1}\right)$
**Output:** weights $\omega$
**while** not converged **do**
    Call dataloader with a random timestep $t$.
    **for** timestep $t \leftarrow t$ to $t+1$ **do**
        $e_1 \leftarrow \sum_{m=1}^M \|\mathcal{D}(s^m(\mathbf{y}_t^m)) - \boldsymbol{x}_t\|_2^2$ according to Eq. 6
        $e_2 \leftarrow \mathcal{L}_{f_{\boldsymbol{\theta}}}(\mathbf{X}_t) + \mathcal{L}_{e2e}(\hat{\mathbf{X}}_t)$ according to Eq. 6
        $e_t \leftarrow e_1 + e_2$
    **end for**
    $\omega \leftarrow \text{Train}\left(\alpha\text{-MDF}, e_t + e_{t+1}\right)$
**end while**
**return** $\omega$

---

