# OpenReview forum: "$\alpha$-MDF: An Attention-based Multimodal Differentiable Filter for Robot State Estimation"
_robot-learning.org/CoRL/2023/Conference — CoRL 2023 Poster_

### Official Review · Reviewer_X85h · 2023-07-17

**Confidence:** 3
**Originality:** Good
**Technical Quality:** Good
**Clarity Of Presentation:** Good
**Impact:** 3

**Recommendation:**

Weak Accept: I recommend accepting the paper, but will not argue for my recommendation if the majority of other reviewers have a different opinion.

**Review:**

The paper presents the α-MDF framework, an attention-based multimodal differentiable filter for improved robot state estimation. The proposed approach leverages attention mechanisms, commonly used in transformer-style neural networks, to learn multimodal latent representations. It substitutes the traditional Kalman gain with a neural attention mechanism for filtering with multimodal observations. This allows the model to learn a highly specialized and task-specific gain mechanism.

The α-MDF framework consists of a transformer process model, sensor encoders, and an attention gain update model. Each module is learnable and operates in the latent space, learning high-dimensional representations of system dynamics and capturing intricate nonlinear relationships. The modular architecture of α-MDF enables it to handle missing sensor modalities, thus improving the model's resilience and robustness. The paper is well-written and well-structured, providing a clear and detailed explanation of the proposed α-MDF framework.

The authors have empirically validated the α-MDF framework on various robot state estimation tasks in real-world settings and simulations. The results demonstrate that α-MDF significantly reduces state estimation errors with nearly a 4-fold improvement compared to current state-of-the-art sensor fusion strategies for rigid body robots. In addition, it outperforms differentiable filter baselines by up to 45% in soft robot modeling tasks.

However, the paper can be further improved in the following areas:
1. The paper has a few typographical errors and requires another pass of proofreading. Some examples are as follows:
a. On page 6, in the last paragraph, the l in "dPF-M-Lrn" is in uppercase. However, in all other instances in the paper, l is written in lowercase.
b. The description of Figure 1 says "Filters" instead of "Filter".

2. The paper lacks a comprehensive comparison in terms of computational complexity with current state-of-the-art models.

3. The paper lacks details on the computational resources used for the experimental setup.

4. It would be beneficial for the paper if the authors could provide detailed sensitivity analysis and ablation studies to demonstrate the effectiveness of the different components of the proposed framework.

5. The authors should provide more details and guidance on hyper-parameter tuning and its impact on the performance of the model.

6. The authors have mentioned limitations at the end of the paper. It would be great if they could provide more insights into how they intend to address these limitations and the scope of future research directions for their work.

**Quality Of The Limitations Section:**

Additional details required

**Questions For Rebuttal:**

It would be great if the authors could address the areas of improvement mentioned in the above section.

**Robotics Focus:**

Relevant but unlikely to deploy to hardware in near future

**Summary Of Paper:**

The paper introduces an attention-based multimodal differentiable filter (α-MDF), which combines differentiable filters with attention mechanisms from transformer models for improved robot state estimation. The α-MDF framework extends the capabilities of traditional DFs by replacing conventional Kalman gain with a neural attention mechanism, effectively generating specialized, context-dependent gains that can efficiently merge multiple input modalities and observed variables. It operates in the latent space, learning high-dimensional representations of system dynamics and capturing complex nonlinear relationships. The paper empirically validates the α-MDF framework on various robot state estimation tasks. The results demonstrate reductions in state estimation errors, with nearly 4-fold improvements compared to other sensor fusion strategies for rigid body robots. Additionally, α-MDF outperforms differentiable filter baselines by up to 45% in soft robot modeling tasks.

**Summary Of Recommendation:**

The paper presents an attention-based multimodal differentiable filter (α-MDF) for robot state estimation. The proposal to use attention mechanisms to replace the traditional Kalman gain is an innovative approach, and the empirical results show promising improvements over existing state-of-the-art methods on various tasks. The writing is clear, and the structure is coherent, making the content easy to follow. However, the paper can be further improved in some areas. It has a few typographical errors and thus requires another pass of proofreading. The authors should provide a comprehensive comparison in terms of computational complexity with other models. It would be beneficial for the authors to provide details on the computational resources used for the experimental setup, detailed sensitivity analysis, ablation studies, details on hyper-parameter tuning, more insights into how they intend to address the limitations, and the scope of future research directions for their work.

---

### Official Review · Reviewer_zXri · 2023-07-20

**Confidence:** 4
**Originality:** Very Good
**Technical Quality:** Good
**Clarity Of Presentation:** Good
**Impact:** 3

**Recommendation:**

Weak Accept: I recommend accepting the paper, but will not argue for my recommendation if the majority of other reviewers have a different opinion.

**Review:**

## Strengths
1. The proposed method operates on the latent space, which opens up the possibility of fusing sensors where the measurement model is obscure (e.x. RGB images).
2. The attention-based gain enables the filter to adjust the gain even when features projected from the measurements live in different spaces.
3. The proposed method was evaluated on 3 distinct applications, and all obtained better results than the baselines.
4. Figure 5 (b) shows interesting qualitative results on the proposed attention value. The network is able to reduce the attention value when a specific sensor modality is disabled.


## Weaknesses
1. The proposed method seems heavily relying on vision sensors. From the ablation study, the IMU doesn’t seem to add much value to the system.
2. The organization of the paper can be improved. Currently, it’s a bit difficult to get a sense of the experimental setup without heavily referring to the appendix.
3. In section 3.1, lines 134-137, the paper states that the traditional Kalman gain is designed to handle a single observation space. I believe this is a bit inaccurate and an abuse of notation. How is the observation space defined here? The traditional Kalman filter does not restrict the measurements to be in the same measurement space (ex., We can simultaneously use velocity measurements and angular velocity measurements to correct the filter.) However, it is required that the function that maps from the state space to the measurement space be explicit. (As a result, some sensor measurements like images or deep-learned features are unable to be used in the formulation directly.) The proposed attention gain relaxes the need for an explicit measurement model.
4. Although mentioned in the limitation section, there’re no experiments demonstrating how well the proposed method works in cases of concept drifts. What happens if the surrounding environment changes drastically for the tensegrity robot experiments?


**Quality Of The Limitations Section:**

Limitations are addressed clearly

**Questions For Rebuttal:**

Issues
1. I might have missed something, but for the manipulation task, why do joint angles need to be estimated as a state? Especially the joint angle measurements are used as an input to the network. The encoder measurements are usually fairly accurate already. I’m confused about why this is needed.
2. In addition to the above point, the end-effector position can be computed through the forward kinematics function, although the kinematic modeling can sometimes be inaccurate. I wonder how the performance of the proposed method compares to pure forward kinematics.
3. Since one of the main contributions of the proposed method is the attention gain, I’m interested to see how the proposed method works compared to a non-learning Kalman filter in cases where a traditional formulation is feasible. The Kalman gain is supposed to be the optimal gain under the Gaussian assumption and linear models. I wonder if the proposed attention gain can achieve similar optimal results when the problem is less complicated and well-constrained.
4. Just out of curiosity, how are the sensors disabled for the experiment shown in Figure 5 (b)? It looks like the sensors were disabled on the fly during continuous inferencing. Was it done by setting the specific input matrix/vector to zero?
5. In Kitti odometry benchmark, it’s a common practice to estimate the 6D motion instead of x, y, and yaw, as the car might go up and down hills. I’m curious to learn how the proposed method works when the search space becomes larger.


**Robotics Focus:**

Sufficient demonstration on hardware

**Summary Of Paper:**

This paper proposes a differentiable Bayesian framework for robot state estimation. The key contributions include attention gain with a learnable queue vector and the formulation of latent space filtering. The proposed method is formulated in an ensemble fashion, where multiple states are propagated and corrected at once to fuse for the best result. It is evaluated on 3 different robotic applications, including visual odometry, manipulation state estimation, and soft robotic state estimation. The experiments show the proposed method achieves higher accuracy than the baselines.


**Summary Of Recommendation:**

Overall, this paper makes the following two contributions: 1) a differentiable Bayesian filter that operates on the latent feature space. 2) A attention-based gain module that allows fusions of features on different spaces. The experimental results show the proposed method achieves higher accuracy than the baselines. As a result, I’m suggesting a weak accept.

---

### Official Review · Reviewer_UVgt · 2023-08-02

**Confidence:** 3
**Originality:** Good
**Technical Quality:** Very Good
**Clarity Of Presentation:** Very Good
**Impact:** 3

**Recommendation:**

Weak Accept: I recommend accepting the paper, but will not argue for my recommendation if the majority of other reviewers have a different opinion.

**Review:**

The paper is generally well written and clear in its presentation of concepts. While the use of attention mechanism is common in many recent machine learning literature, its use in replacing the gain matrix in the filtering process is a simple yet clever idea.

Strengths:
1. The introduction of the attention mechanism to replace traditional gains is refreshing. It allows efficient context-dependent prediction of the gain directly (during inference) without explicit understanding of the system-specifics (e.g., observation matrix, noise covariance matrix) since the system is modeled as a neural network in a data-driven manner.

2. Additionally, the design of the attention mechanism which allows variable input length means that for different tasks and system of interest, the operators have the flexibility to consider different input modalities using the same architecture. Such a consideration is not viable for the baselines presented in the paper, and an additional fusion strategy is necessary.

3. The empirical results presented in the paper shows that the approach is vastly superior to existing state-of-the-art and the simplicity of the proposed neural network model makes it even more attractive.

Weakness:
1. While the author considers operating within a latent space as a primary contribution of the paper (contribution 2), it is hard to claim so. Arguably, every neural network model, whether it is an expansion from a low to high dimensional space or a compression from high to low dimensional space, works with a latent space and the use of neural networks to model nonlinear relationships is not unique to this work.

2. Little information is given for the transformer-style neural network $f_\theta$, presented in the prediction step. Additional information can be provided to allow a better understanding of its implementation especially when $f_\theta$ takes in not only a sequence of previous states $\mathbf{X}_{t|t-N:t-1}$ but also an optional action $\mathbf{a}_t$ which could be of a different dimension as seen in the experiments presented. While $f_\theta$ is not a main focus in the paper, the prediction step remains a crucial component of the filtering process and its implementation should be clearly presented. The authors can provide a more detailed implementation in the appendix if the length of the paper does not permit its inclusion.

**Quality Of The Limitations Section:**

Additional details required

**Questions For Rebuttal:**

1. The final loss function proposed is the summation of three different losses, namely the state transition loss ($L_{f_\theta}$), end-to-end loss ($L_{e2e}$), and latent observation conditioning loss ($L_s$). L2 loss between the decoded latent and the ground truth state, $\mathbf{x}_t$ is used for the three sub-losses. While the first two losses are sensible (i.e., matching the decoded prediction with the ground-truth and matching the decoded estimation with the ground-truth), the final sub-loss seems perplexing. By using the loss $L_s = ||\mathcal{D}(s^m(\mathbf{y}_t^m)) - \mathbf{x}_t||_2^2$, the neural network attempts to match the (decoded) latent observation $\mathcal{D}(s^m(\mathbf{y}_t^m))$ to the ground-truth state (e.g., matching a latent state derived from depth image to the ground-truth state). Is such a loss possible? For example, in the multimodal manipulation task when $\mathbf{y}^4$ is the force/torque sensor reading. Does the observation necessarily contain sufficient information to derive ground-truth state?

2. In the attention gain module, a matrix $\tilde{\mathbf{M}}$ is introduced to retain only the diagonal elements of the $d_x \times d_x$ attention map in support of the argument that "within each latent vector, every index is probabilistically independent, and index $i$ of a latent state should only consider index $i$ of each latent observation". Is there any evidence or prior works to support this claim? The claim that "$\tilde{\mathbf{M}}$ can be disabled based on different input sensor modalities, thus improving the model's resilience to missing modalities" also seem unconvincing as it is difficult to relate how the removal of $\tilde{\mathbf{M}}$ makes the model resilient to missing modalities.

3. In the multimodal manipulation task, $\alpha$-MDF outperforms baselines dEFK and DPF in the estimation of the end-effector positions. It seems that only the RGB modality is used for dEKF since the DF baselines only take one modality as stated and the experiment (Table 1) does not consider sensor fusion techniques. Does $\alpha$-MDF only consider the RGB modality in this case? In the  subsection on task setup and data, it seems to imply that $\alpha$-MDF takes in more than just the RGB observations. If that is the case, then the comparison of results will not be a fair one since the $\alpha$-MDF takes in additional information when making the estimation.

**Robotics Focus:**

Sufficient demonstration on hardware

**Summary Of Paper:**

The paper proposes a data-driven filtering process which eliminates the need for an explicit analytical model of the system of interest. The attention-based Multimodal Differentiable Filter ($\alpha$-MDF) replaces the traditional gain matrix in the filtering process with an attention unit and a learnable query, which propagates through both the latent state and observations to output an attention gain used to update the estimate of the (latent) state. The flexibility of the attention mechanism also allows efficient fusion of multimodal observations which the authors show empirically, leads to superior state estimation performances compared to recent state-of-the-art.

**Summary Of Recommendation:**

While there are some issues that needs to be cleared up, the general concepts are sensible and the overall idea is interesting. The superior performances seen in the experiments further show that the paper is deserving.

I am satisfied with the rebuttal by the authors.

---

### Author Response · Authors · 2023-08-10
**Reply to All the Reviewers**

We would like to express our sincere appreciation for the time and effort all the reviewers invested in the evaluation of our work. We performed 6 new experiments suggested by the reviewers which are reported in sections Appendix B.1.2, B.1.3, B.2.2, B.2.3, B.3.2, and C.1. We are glad they acknowledge the strengths of our work, finding the idea to be simple yet clever (UVgt), refreshing (UVgt) and innovative (X85h), as well as acknowledging the contribution of proposing attention-based gains (UVgt, zXri) which we believe will lead to family of filtering algorithms. We are also glad the reviewers agree that our method achieves superior/better results than state-of-the-art and other baselines (UVgt, zXri), outperforming other differentiable filters by nearly 4-fold or up to 45% on different tasks (X85h). Additionally, we are pleased that reviewer UVgt recognizes the flexibility of our proposed operator which goes beyond by other filtering approaches. We would like to highlight that the experiments recommended by the reviewers are a great complement to the paper and that we have performed *all of these suggested experiments*.

Next, we will address the reviewer comments point-by-point.

---

### Decision · Program_Chairs · 2023-08-30

**Decision:**

Accept (Poster)

**Comment:**

This paper presents an attention-based methodology for robot estimation that eliminates the need for an explicit analytical model of the system.

The reviewers have found that the approach is interesting and valid, and also the articila is well written. The rebuttal has better clarified on the points the reviewer pointed out for improvement.